# Algal symbiont diversity in *Acropora muricata* from the extreme reef of Bouraké associated with resistance to coral bleaching

Cinzia Alessi[1,2]*, Hugues Lemonnier[1,2], Emma F. Camp[3], Nelly Wabete[1], Claude Payri[1,2], Riccardo Rodolfo Metalpa[1,2,4]

1 ENTROPIE, IRD, Université de la Réunion, CNRS, IFREMER, Université de Nouvelle-Calédonie, Nouméa, New Caledonia, 2 Laboratoire d'Excellence CORAIL, ENTROPIE (UMR9220), IRD, Nouméa, New Caledonia, 3 Climate Change Cluster, University of Technology, Ultimo, NSW, Australia, 4 Labex ICONA International CO2 Natural Analogues Network, Shimoda, Japan

* cinzia.alessi@ird.fr

**Data Availability Statement:** All raw sequence data as fastq read files are accessible under NCBI Sequence Read Archive (SRA), under NCBI's

## Abstract

Widespread coral bleaching has generally been linked to high water temperatures at larger geographic scales. However, the bleaching response can be highly variable among individual of the same species, between different species, and across localities; what causes this variability remains unresolved. Here, we tracked bleached and non-bleached colonies of *Acropora muricata* to see if they recovered or died following a stress event inside the semi-enclosed lagoon of Bouraké (New Caledonia), where corals are long-term acclimatized to extreme conditions of temperature, pH and dissolved oxygen, and at a nearby control reef where conditions are more benign. We describe Symbiodiniaceae community changes based on next-generation sequencing of the ITS2 marker, metabolic responses, and energetic reserve measures (12 physiological traits evaluated) during the La Niña warm and rainy summer in 2021. Widespread coral bleaching (score 1 and 2 on the coral colour health chart) was observed only in Bouraké, likely due to the combination of the high temperatures (up to 32°C) and heavy rain. All colonies (i.e., Bouraké and reference site) associated predominantly with Symbiodinaceae from the genera *Cladocopium*. Unbleached colonies in Bouraké had a specific ITS2-type profile (proxies for Symbiodiniaceae genotypes), while the bleached colonies in Bouraké had the same ITS2-type profile of the reef control colonies during the stress event. After four months, the few bleached colonies that survived in Bouraké (B2) acquired the same ITS2 type profiles of the unbleached colonies in Bouraké. In terms of physiological performances, all bleached corals showed metabolic depression (e.g., $P_{gross}$ and $R_{dark}$). In contrast, unbleached colonies in Bouraké maintained higher metabolic rates and energetic reserves compared to control corals. Our study suggests that *Acropora muricata* enhanced their resistance to bleaching thanks to specific Symbiodiniaceae associations, while energetic reserves may increase their resilience after stress.

BioProject: PRJNA1020910), and data are available at DOI: 10.5061/dryad.05qfttf8z.

**Funding:** CA was supported by a PhD fellowship from Labex CORAIL and IFREMER. The research was supported by the French Ministry of Foreign Affairs, Fonds Pacifique project "SuperCoraux" #6614A1. Sequencing costs and the contribution of EFC to the project were supported by a CPDRF grant awarded to EFC. Include this sentence at the end of your statement: The funders had no role in study design, data collection and analysis, decision to publish, or preparation of the manuscript.

**Competing interests:** The authors have declared that no competing interests exist.

# Introduction

Coral reefs are one of the most biodiverse and complex ecosystems on the planet that provide ecosystem goods and services to support healthy marine ecosystems [1]. However, coral reefs are degrading worldwide because of global climate change [2–4]. Elevated seawater temperatures induced by anthropogenic global warming are primarily responsible for increasing the frequency of bleaching events, which have a high likelihood of happening during one of the El Niño–Southern Oscillation phases (ENSO) [5]. Coral bleaching [6] is characterized by the loss of algal symbionts (and/or pigmentation) in response to local environmental conditions, causing variations in the coral color [7,8]. Corals experiencing bleaching are particularly vulnerable since they cannot rely on the Symbiodiniaceae photosynthetic carbon for their daily metabolic requirements [9,10]. Some corals can rely on the acquisition of heterotrophy carbon, thus capturing particles and/or assimilation of dissolved inorganic and organic compounds [11,12]. In general, excess in autotrophic and heterotrophic carbon can be stored in the host as energy reserves like lipids that normally represent 10–40% of total biomass [13–15]. However, autotrophy depletion forces bleached corals to rely on stored lipids, carbohydrates, or protein reserves to satisfy their carbon requirements until bleaching recovery occurs [16]. Bleached corals are unstable and their likelihood of survival depends on bleaching severity, duration of the event, nutritional plasticity (e.g., ability to shift between heterotrophy and autotrophy), and energy reserves achieved prior to bleaching [17]. In some cases, surviving corals can rapidly adjust their thermal tolerance within and between host generations by shuffling their microalgal symbiont with a more stress-tolerant species [18–20]. Indeed, some symbiont species such as *Durusdinium trenchii* have been described as more resilient than others under elevated temperature, even if less efficient in terms of carbon translocated to the host [21,22]. In other cases, the coral host can adjust its thermal tolerance by expressing stress resistance genes [23,24] and promoting genetic adaptation to local environments [18,24,25]. The bleaching threshold can be interspecific and intraspecific depending on environmental memory and thermal tolerance of both the coral host and algal symbiont [8,26,27], leaving the exact triggers of bleaching still unsolved. Recently, studies are focusing on natural extreme reef environments inhabited by corals, such as the sheltered bay of Palau [28–30], reef tide pools on Kimberly [31,32], semi-enclosed lagoons [33–35], and mangrove environments [20,36,37] to understand the phenotypic plasticity, and the remarkable resilience of corals inhabiting such natural laboratories to chronic and acute environmental disturbances [32,38]. Notwithstanding such resilience, during a warm and rainy period associated with La Niña (January 2021) a partial bleaching and mortality of corals, mostly Acroporidae, were observed in the semi-enclosed lagoon of Bouraké in New Caledonia, but not in other localities of New Caledonia. A similar bleaching and mortality event during an extreme wet period was recently documented for a mangrove-coral location on the Great Barrier Reef [39]. Bouraké shelters a healthy reef that is chronically exposed to daily variations in pH, temperature, dissolved oxygen, and salinity [34]. This severe bleaching event was the first to be recorded after 2016, when a mass coral bleaching event impacted entire reefs in New Caledonia [40], while only a mild paling of corals was observed in Bouraké (Rodolfo-Metalpa, *in situ* observations). There is increasing work being undertaken to reveal diverse strategies undertaken by these corals to survive under chronic disturbances [20,33,38,41,42], yet what intraspecific stress-induced physiological response of extreme corals (i.e., corals that live at the edge of their environmental limits [43,44], occur during a stress event remains entirely unexplored.

The aim of this study was therefore to explore potential mechanisms underlying the physiological plasticity of corals that live in extreme and fluctuating conditions during a period of acute stress (i.e., elevated temperatures and La Niña in 2021). We took advantage of a natural

bleaching event to investigate possible shifts in the endosymbiont community and key physiological indicators of metabolic response in *Acropora muricata* comparing healthy and bleached corals from Bouraké with healthy colonies from a reference. To assess the physiological responses during the stress event we quantified algal symbiont physiology (cell density, chlorophyll a concentration, photosynthetic efficiency (e.g., Maximum quantum yield and the maximum rETR), Symbiodiniaceae photosynthesis and holobiont respiration) and coral host energy reserves during the bleaching event and four months post bleaching. We also investigated symbiont community dynamics using next-generation sequencing of the ITS2 marker for both Bouraké and reference corals at the two time points.

## Methods

### Study sites and environmental parameters

Two study sites were selected, B2 in Bouraké and the reference site R1, which is situated 1.5 km from Bouraké [33] (S1 Fig, see supporting information). Seawater temperature was measured at both sites (ca. 2 m depth) from December 2020 to May 2021, using HOBO water temperature Pro V2 sensors set at 10-minutes logging intervals. At both sites, the physio-chemical environmental variability, coral species composition, and coral species abundances have been monitored since 2016 and described by [34]. Daily variations in pH and oxygen (DO) in Bouraké are connected to the tidal cycle, and their regular oscillations have been reported in several studies [33–35,45] suggesting variations are consistent over time. Studies reported a chronic range in $pH_T$ from 8.06 to 7.23, and regular fluctuations in DO between 7.10 and 2.28 (mg $L^{-1}$ $h^{-1}$) [34,35]. Therefore, pH and DO were not monitored during the following study. Data on rain were taken by Meteo France at the nearby Bouraké station.

### Colonies selection, coral bleaching and mortality assessments

Study sites were visited on the 6th December 2020, and no visual signs of bleaching were observed. On revisiting the sites in early January 2021, coral bleaching and mortality were observed in Bouraké (S1 Fig), but not at the reference site. Additional observations were made in early February at the reference site R1, and no bleaching was observed. Coral bleaching and mortality were assessed on the 18th of January 2021 (T1, Bleaching) and on the 5th May 2021 (T2, Post bleaching) in B2 and R1. At each site, three 10 m permanent transects, each separated by at least 5 m, were set out. A 50 x 50 cm PVC quadrat was positioned on transects at 1 m intervals and photographed using an underwater camera (Canon G16 with Fantasea underwater case). Coral genus and health status were assessed on each photo using a Photo Quad with 40 random points in each quadrat. We identified four main categories: healthy coral cover (i.e., score 3–6 of the coral colour health chart); bleached coral cover (i.e., score 1 and 2); recently dead coral cover (i.e., dead corals with visible white skeletons not covered by algae); and dead coral cover (i.e., dead corals with skeletons covered by algae). Based on the species list previously made at the two sites [34], the main coral species affected by bleaching and mortality were identified in the field. *Acropora muricata* was selected as the model species to track physiological changes during the bleaching and post-bleaching for two reasons: 1) it was one of the most affected species; 2) the Symbiodiniaceae community of this species was already described for Bouraké's population by Camp et al. [33]. In Bouraké (B2), 15 pigmented colonies (score 5 or 6 of the coral colour chart, hereafter defined as Bouraké Zoox, BZ) and 15 bleached colonies (score 1 or 2 of the coral colour chart, hereafter Bouraké Bleach, BB) of *A. muricata* were tagged in January (T1). In contrast, at the reference site R2 only nine pigmented colonies belonging to the same species were found and labelled (hereafter Reference Zoox, RZ). Colonies were at the same depth, with a similar diameter (40 to 50 cm), and with no signs

of mortality at the time of the colony selection. Four fragments (4–5 cm in length each) were sampled from the centre of each colony and from each study site at both T1 and T2. Fragments from each colony were kept in individual zip bags containing native seawater and transported to the nearby field lab (Research station ADECAL-IFREMER in St. Vincent). One fragment per colony was preserved in ethanol for the Symbiodiniaceae genotyping; one fragment was used to measure photosynthetic efficiency, and then it was frozen at -80˚C and stored for further analyses on lipids, carbohydrates, and biomass; one fragment was used to measure metabolic rates, and then chlorophyll, Symbiodiniaceae and total proteins contents; while the last fragment was weighed and transplanted back to the sampling sites to assess the corals growth rate (see below).

## Growth rate

At each time point, T1 and T2, and at each site, B2 and R1, one fragment from each colony was transplanted at their original site in a common garden experiment, and its growth rate was measured after three weeks. However, the number of replicates of the BB category was reduced from 15 to 3 in T2 because of the high mortality of bleached colonies during and after the bleaching. The buoyant weight technique [46] was used to weigh fragments in a Sartorius ENTRIS 224i-1S electronic balance (readability 0.1 mg) in seawater of known density which was calculated from temperature and salinity. Then, each fragment was mounted on an individual labelled PVC support, weighed again, and secured on concrete blocks at their original site of collection. Three weeks later, nubbins were collected and weighed. Dry skeleton weight was calculated using the density of pure aragonite (2.94 g cm$^{-3}$), and growth rate was calculated as the change in dry weight of the coral skeleton between the initial and the final weight and expressed in mg g$^{-1}$ d$^{-1}$.

## Photosynthetic efficiency, Symbiodiniaceae photosynthesis, and respiration

One fragment from each colony and from each site was dark adapted for 30 minutes in native seawater from each respective study site, and at a room temperature of 26˚C. For each measurement the DIVING-PAM optical fiber was placed perpendicularly to the coral surface, ca. 1 cm below the axial corallite, and at a fixed distance of 5 mm to the fragment's surface using a plastic spacer. Rapid Light Curves (RLCs) were applied using the internal program of the DIVING-PAM (Walz, Germany) that provides a sequence of nine-light steps with light intensities increasing from 5 to 1800 µmol photons m$^{-2}$ s$^{-1}$ (settings: measuring light 8, saturating intensity 8, saturating width 0.8 s, gain 3 and damping 2). Each illumination period lasted 10 s and finished with a saturating pulse that measured the effective quantum yield ($\Delta F/Fm'$). Maximum quantum yield (Yield$_{max}$) was measured at the first light step. The maximum rETR (rETR$_{max}$) was calculated by multiplying the effective quantum yield by each light intensity. After measurements, coral fragments were frozen and then used to measure lipids, carbohydrates, and biomass (see below).

Another coral fragment was used to measure the net coral photosynthesis (P$_n$) and dark respiration (R$_{dark}$) rates. Fragments were incubated in 100 ml glass beakers filled with native filtered seawater (1 µm) constantly mixed using magnetic stir bars. Beakers were hermetically closed using transparent plastic film to avoid any oxygen exchange with air. Each beaker was equipped with an oxygen sensor spot (SP-PSt6-NAU, PreSens, Germany), and oxygen concentration was measured with a polymer optical fiber and Fibox 4 (PreSens) at the beginning and at the end of the light and dark incubations, respectively [35,47]. The incubation temperature represented the mean value (± SD) measured in the field by the deployed HOBO temperature

loggers (ca. $29.7 \pm 1.13°C$ at T1, from January $1^{st}$ to January $17^{th}$, and ca $26 \pm 1.21°C$ at T2, from April $20^{th}$ to May $1^{st}$). Temperatures were kept stable using a temperature-controlled water bath. Two control beakers containing only filtered native seawater were run in parallel to account for the background oxygen change by microorganisms in the seawater. Light irradiance was provided by LED lights (Mitras LX6100, GHL Germany) at 190 μmol photons $m^{-2}$ $s^{-1}$ and measured with LI-193 Spherical Quantum Sensor (LI-COR, USA). After 50 minutes of incubation in the light, $P_n$ rates were measured. Then, the light was switched off, and the corals were given 15 minutes to acclimate to darkness before measuring their respiration rates for a further 50 minutes. $P_n$ and $R_{dark}$ rates were corrected for background oxygen changes measured in the control beakers and scaled for the water volume. Gross photosynthetic ($P_{gross}$) rates were determined by adding the dark respiration to $P_n$, while $P_g$:R was calculated by multiplying $P_{gross}$ for 12h and $R_{dark}$ for 24 h. All rates were normalized for surface area, which was obtained with the wax dipping method [48,49].

## Biomass, lipids, carbohydrates, and Symbiodiniaceae parameters

One coral fragment 2–3 cm in length from each tagged colony was freeze-dried and weighed. The surface areas were measured using the aluminium foil technique [50], and the skeleton was then ground to powder in an agate mortar [51]. Tissue biomass was measured by drying a sub-sample of the powder, pooling skeleton, animal tissue, and algal endosymbionts for 24 h at 60°C, followed by burning for 5 h at 450°C. The difference between dry and burned weight was the ash-free dry weight, which was standardized to the surface area of the sub-sample to obtain the biomass [52,53]. Another coral powdered sub-sample was used to determine soluble lipids in the pool of skeleton, animal tissue, and algal endosymbionts. According to Folch [54], the coral powder was weighed and then mixed with 1 ml of dichloromethane: methanol solution (2:1 ratio), followed by two successive cycles of centrifugation at 3500 g for extraction in 0.75% NaCl. The lipid extract was dried to constant weight and standardized to the biomass. Carbohydrates were also extracted using a sub-sample of the same coral powder. Milli-Q water was added to the ground coral sub-sample after being weighed, and the resulting slurry was centrifuged twice (5000 g, for 10 min) to separate the animal tissue from the rest of the sample. Carbohydrates were extracted following Bove and Bauman [50,55] and Masuko et al., [56] Lipids and carbohydrates were normalized to the biomass to account for the lack of Symbiodiniaceae in bleached colonies at T1.

The tissue of the fragments used for photosynthetic measurements was removed using an air pick in 22 ml of filtered seawater (Whatman GF/F) to quantify the Symbiodiniaceae density, chlorophyll content, and protein content. The slurry was then homogenized using a grinder potter, and a sub-sample (2 ml) was taken for the determination of the symbiont density per surface area. For that, Symbiodiniaceae were counted in six replicates for each coral on a hemocytometer (Neubauer) under a microscope. Chlorophyll content was measured using 10 ml of the coral slurry centrifuged at 6000 g for 10 min at 4°C. The supernatant was discarded, and 10 ml of pure acetone added to the pellet and left for 24 h in darkness at 4°C. Absorbance was measured at 630, 663, and 750 nm using a spectrophotometer (Evolution 201 UV-Visible, Thermo Fisher Scientific). Total chlorophyll ($a + c_2$) concentrations were calculated using the equation of Jeffrey and Humphrey [57]. Proteins were extracted by mixing 500 μl subsample of the coral slurry (animal tissue and algal endosymbiont) with 500 μl of NaOH 0.5 N. Then, samples were incubated in the autoclave for 5 h at 60°C. Soluble proteins were determined by adding a sample aliquot (25 μl) to a BCA Protein Assay Kit (Interchim) and using bovine serum albumin as a standard. Absorbance was set at 562 nm and soluble proteins were measured using a spectrophotometer (Evolution 201 UV-Visible, Thermo Fisher Scientific). Total chlorophyll and soluble proteins were normalized to surface unit.

## Symbiodiniaceae DNA extraction, PCR amplification and sequencing

Coral fragments of ca. 2 cm length were preserved in absolute ethanol for genotyping of Symbiodiniaceae based on amplicon sequencing of the Ribosomal Internal Transcribed Spacer 2 (ITS2). Total coral holobiont DNA (i.e., Symbiodiniaceae, polyp and associated microorganisms DNAs) was extracted using a 2% CTAB-based protocol adapted from [58]. The quantity and quality of extracted DNA were checked using a NanoDrop 2000 spectrophotometer (Thermo Fisher Scientific, MA). Extracted DNA was then diluted to a range of 30–70 ng μL$^{-1}$ for PCR amplification. The Symbiodiniaceae nuclear DNA ribosomal internal transcribed spacer (ITS2) region was amplified with the forward primer ITS2-DINO [5′–TCGTCGGCAG CGTCAGATGTGTATAAGAGACAGGTGAATTGCAGAACTCCGTG–3′] [59] and reverse primer ITS2Rev2 [5′–GTCTCGTGGGCTCGGAGATGTGTATAAGAGACAGCCTCCGCTTACTTATAT GCTT–3′] [60]. The underlined segments represent Illumina adapter overhangs (Illumina, San Diego, CA, USA). The PCRs were conducted in 25 μL reactions using 12.5 μL of AmpliTaq 360 Master Mix, 1 μL of each 10 μM primer mix, 1 μL of 360 GC Enhancer, 2 μL of DNA template and DNAse-free water to adjust the reaction volume. The amplification cycle was set and adjusted from Arif et al. [61] as follows: 94˚C for 15 min; 35 cycles each at 95˚C for 30 s, 49˚C for 1 min, and 72˚C for 30 s; and a final extension at 72˚C for 10 min. To check for amplification success, 3 μL of each PCR product were run on a 1% agarose gel. The resulting amplicons were sequenced using the Illumina MiSeq platform (2 x 300 bp) (Australian Genome Research Facility, Victoria, Australia, average sequencing depth: 97252). Returned demultiplexed FASTQ files were analyzed via the SymPortal analytical framework [62]. The SymPortal framework predicts ITS2-type profiles from specific sets of defining intragenomic ITS2 sequence variants (DIVs) based on genetically differentiated Symbiodiniaceae taxa. Quality control was assessed using Mother 1.39.5 [63], BLASTC suite of executables [64] and minimum entropy decomposition [65] to predict Symbiodiniaceae taxa from the ITS2 marker.

## Statistical analyses

Seawater temperature daily means were compared between study sites (Bouraké and R1) using one-way ANOVA. Multivariate two-way PERMANOVAs (9999 permutations) were performed to test changes in relative abundance in ITS2 sequences and predicted ITS2-type profiles among categories (BZ, BB, RZ) and time points (T1 and T2), and also to test changes in the holobiont physiological traits (biomass, soluble lipids, soluble proteins, carbohydrates, $P_{gross}$, $R_{dark}$, Pg:R chlorophyll, Symbiodiniaceae, $rETR_{max}$, and $Yield_{max}$). The growth rate was not included in the physiological traits because of the limited number of transplanted replicates that survived at T1. Data were normalized, and a square root transformation was applied prior to the construction of the Bray-Curtis and Euclidean distance resemblance matrix for the analyses on Symbiodiniaceae, and physiological profiles, respectively. The PERMDISP test for homogeneity of the dispersion [66] was also done using distances to centroids and with $p$ values obtained using 999 permutations of residuals. PERMANOVA was performed setting categories (three levels: BB, BZ, and RZ) and time (two levels: bleaching (T1) and post-bleaching (T2)) as fixed effects. When significant differences were found, a post hoc pairwise test was run. The major contribution to the variability among categories and time points was suggested by similarity percentage analysis (SIMPER), and data visualization was based on non-parametric multidimensional scaling (nMDS) plot. In addition to the multivariate analysis, differences in the phenotypic traits among categories were also assessed separately for each time point using multivariate one-way PERMANOVAs (9999 permutations) after having checked the homogeneity of variance (PERMDISP, centroids, and $p$-value at 999 permutations). PERMANOVA is a robust non-parametric test that can be applied in balanced and unbalanced designs,

although, in the presence of heterogeneity of variance, it is considered to be reliable in balanced design only [67,68]. PERMANOVA and pairwise analyses were also used on energetical reserves (e.g., carbohydrates, biomass, lipids, and proteins) of the few colonies that survived at T2 between time points. ANOVA was performed with R software (v.3.4.3), whereas PERMANOVA pairwise and SIMPER analyses were performed using Primer software V6. All raw sequence data as fastq read files are accessible under NCBI Sequence Read Archive (SRA), under NCBI's Bio-Project: PRJNA1020910), and data are available at DOI: 10.5061/dryad.05qfttf8z.

## Results

### Environmental temperature and rainfall regime

The average daily temperature from December 2020 to May 2021 were significantly different between sites (one-way ANOVA, $F_{1,362} = 10.966$ p = 0.001) and ranged from 21.37˚C to 32.89˚C in Bouraké (mean ± SD = 27.32 ± 2.02˚C), and from 22.08˚C to 30.95˚C at the reference site R1 (mean ± SD = 26.94 ±1.56˚C) (Fig 1). During the four months of observation, the daily seawater maximum temperature exceeded 30˚C during 75 days in Bouraké and only 12 days in R1. Daily average temperature exceeded 30˚C only in Bouraké, over 14 days between January (bleaching peak) and the end of February 2021.

Concomitantly to the first heat wave in January 2021, New Caledonia experienced three events of exceptionally strong precipitations, which marked the beginning of a period of frequent episodes of rainfall events linked to La Niña, likely the most intense since the year 1981 (source: Meteo France). In addition, during the four months of observation, tropical storm Lucas (3-4/02/2021) and cyclone Niran (5-6/03/2021) hit New Caledonia and passed close to

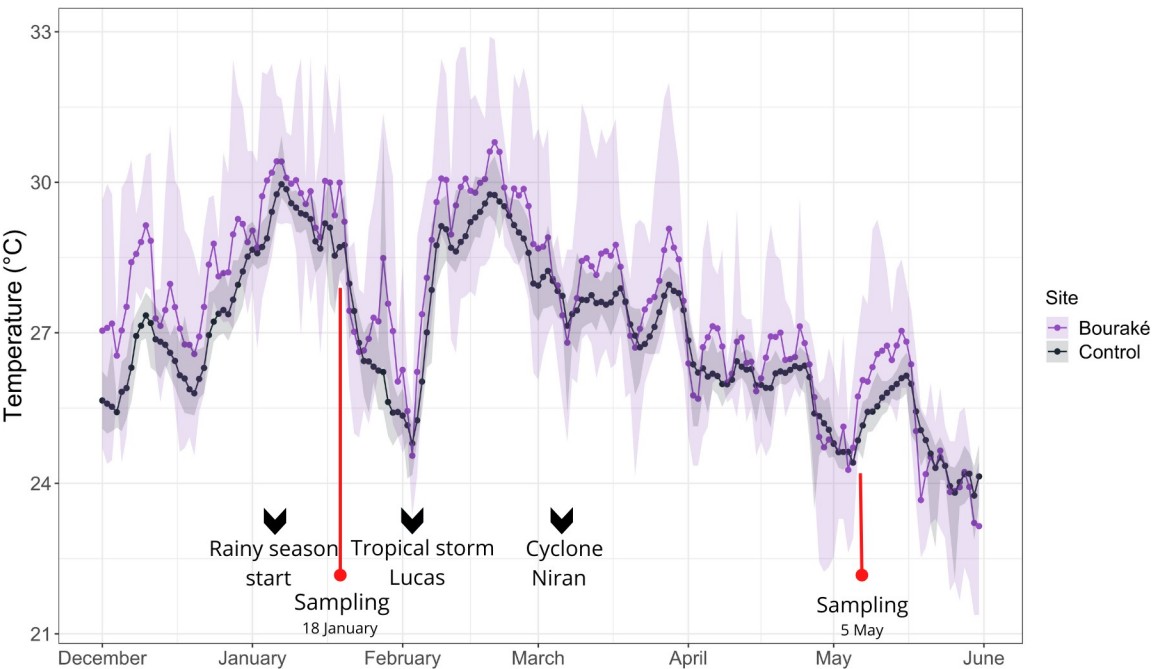

**Fig 1. Temperature profiles for Bouraké and the control reef site.** Daily average temperature (solid line) and daily temperature range (shaded area) measured in Bouraké (B2), and the reference reef site (R1) from the 1st of December 2020 to the 31st of May 2021. Sampling activities were undertaken on the 18th of January (T1) and on the 5th May (T2) 2021. Temperature drops were recorded during periods of intense rain fall, starting from the beginning of January (first 15 days). Episodic heavy rain events were recorded during the passage of the tropical storm Lucas (3rd and 4th February 2021), and the cyclone Niran (4th-5th March 2021).

our study sites. A total of 300 mm of precipitation was measured in the Bouraké region (including the study site and nearby reference site) from the 30th of December 2020 to the 18th of January 2021, which was 5.3-fold higher than the average precipitation measured from 1981–2010 during the same period (S2 Fig).

## Coral bleaching and mortality

In January 2021 (T1), 47.5% of corals in Bouraké were visually healthy (e.g., high pigmentation), while 33.2% were found bleached. The remaining corals were already dead, with 16.2% of them categorized as recently dead (i.e., no signs of polyp extroversion, but the skeleton was still white, and the coral not fully repopulated by epiphytes algae and encrusting organisms), while only 2.9% were recognized as old coral mortality (Fig 2). The most affected coral genus was *Acropora* with 36.82% of the coral cover bleached (out of 85.4% of total cover for this genus). Minor percentages were found for *Montipora* spp. (1% bleached cover out of 10.8% of total coral cover), and less than 1% of bleached colonies for *Porites* spp. and *Pocillopora* spp., which together represented the remaining 3.8% of total coral cover. No coral bleaching was observed at the reference site R1 and, more generally, in the same geographical area.

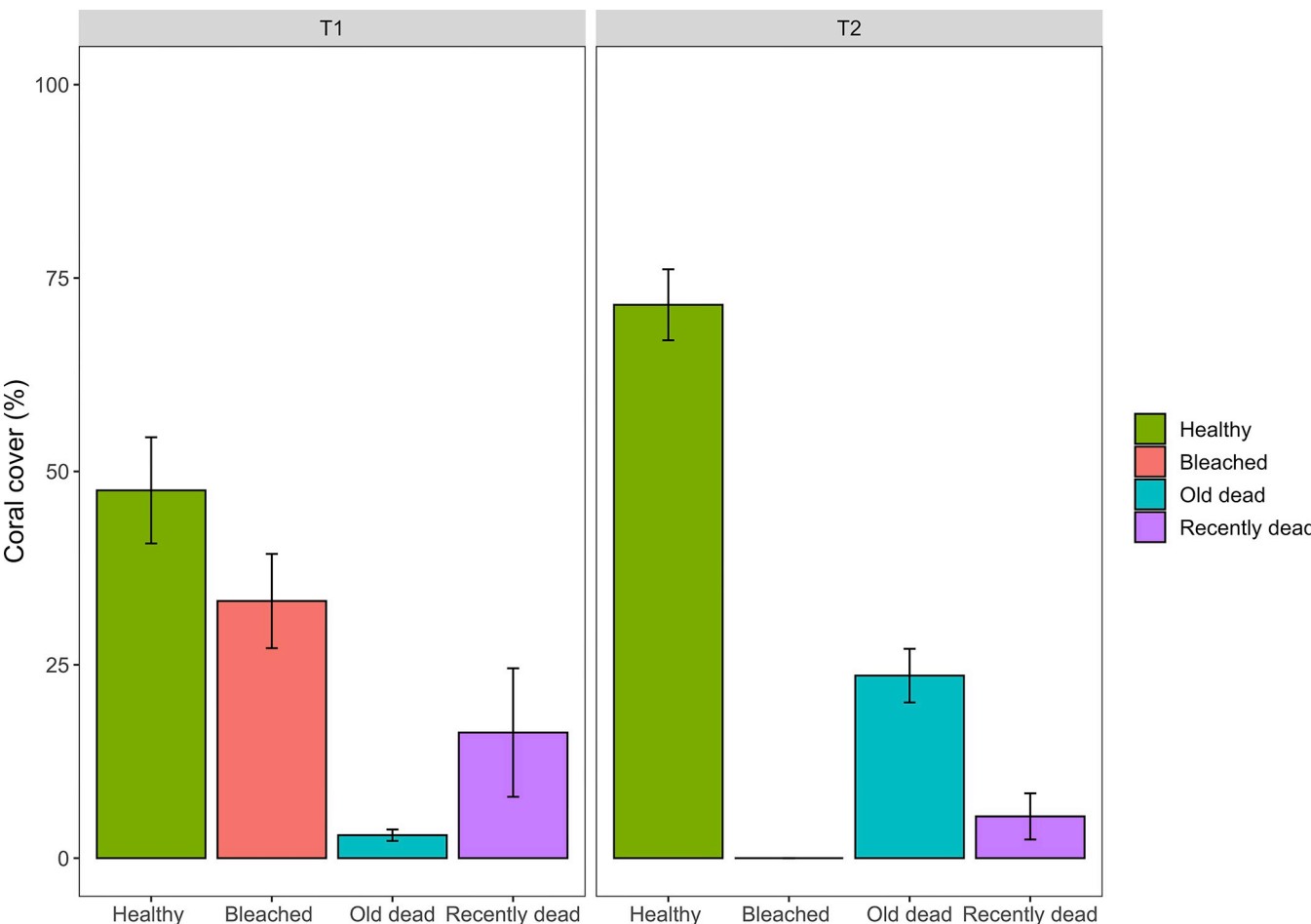

**Fig 2. Coral cover response to bleaching in Bouraké.** The percentages of healthy (score above 3 on the coral color health chart), bleached (score 1 and 2 of the coral color health chart), dead, and recently dead corals in Bouraké (B2), during the bleaching event (January 18th 2021), and four months post bleaching 5th May 2021 (T1, and T2, respectively). No data were presented for the reference site (R1) as no bleached colonies were observed. Values are mean ± SD per transect (n = 3).

In May 2021 (T2), most of the bleached corals recovered, resulting in 71% of corals fully pigmented. However, old coral mortality increased up to 23.5% and recently dead corals decreased to 5.3%, respectively. Concerning the tagged colonies in Bouraké, mortality was very high for the bleached colonies (BB) and little for the unbleached "healthy" ones (BZ). Indeed, only three BB colonies survived, although they had between 20 and 50% partial mortality. In contrast, only 20% of the 15 BZ colonies completely died, while some surviving colonies showed signs of partial mortality (S1 Table). The three surviving BB colonies, during recovery returned their normal pigmentation (e.g., color card from 3 to 5) and increased their Symbiodiniaceae density in T2 (see below). No signs of mortality were recorded on the tagged colonies in R1, although four colonies were lost after the passage of the cyclone Niran.

### Effect of stress on the holobiont physiological traits

Two-way PERMANOVA showed that physiological profiles significantly differed between coral categories, sampling times, and their interaction (Table 1, S4 Fig).

The data plot (nMDS) showed clear separations in all colony categories between times (S3 Fig), which was confirmed by the pairwise analysis (Table 2). At T1, SIMPER analysis suggested Symbiodiniaceae density, $rETR_{max}$, Pg:R and carbohydrates to be the physiological traits that mainly explained the largest variance between BB and the remaining coral categories, while the main differences between RZ and BZ were caused by differences in $Yield_{max}$, carbohydrates, and lipids (Table 2). At T2, SIMPER analysis showed that carbohydrates, proteins, and biomass were the physiological traits of the three surviving BB colonies which mostly explained the difference with RZ (Table 2).

Carbohydrates, biomass, and respiration were the physiological traits contributing most to the variance between BZ and RZ. One-way PERMANOVAs were run separately for T1 and T2, highlighting differences in each physiological trait among categories. Pairwise comparisons revealed that during the bleaching event at T1, all traits in BB were significantly affected compared to BZ and RZ, except for proteins, and biomass where no differences among categories were found (Fig 3; Table 3).

Healthy colonies BZ showed significantly greater $rETR_{max}$ (38%), Yield (20%), chlorophyll (44%), Symbiodiniaceae density (32%), and carbohydrates (65%) when compared to RZ (Fig 3D, 3F, 3G and 3M), but lower growth rate (35%) and $P_{gross}:R_{dark}$ (26%) (Fig 3G and 3N; S3 Table). Interestingly, the BB colonies had similar $Yield_{max}$, carbohydrates, and protein contents compared to RZ (Fig 3F, 3I and 3M), although the bleaching conditions.

Post bleaching, at T2, only some of the initial colonies survived (i.e., n = 3, n = 12, and n = 6 for BB, BZ, and RZ, respectively). Although this resulted in limited replication, the three fully bleached colonies (BB) that survived, totally recovered, showing similarity to the other categories (i.e., BZ and RZ) in all of the physiological traits, except for carbohydrates and protein content, where values were significantly higher compared to RZ (Fig 3; S2 Table). At the same time, a few significant differences in the physiological traits were found between BZ and RZ. Indeed, BZ had lower $P_{gross}:R_{dark}$, but higher Symbiodiniaceae density, protein, carbohydrates,

**Table 1. Two-way PERMANOVA on physiological traits of *Acropora muricata* (biomass, total lipids, total protein, carbohydrates, chlorophyll, $P_{gross}$, $R_{dark}$, $Yield_{max}$, and Pg:R), between coral conditions (BZ, BB, and RZ) and sampling times (T1 and T2).**

| Source | df | SS | MS | Pseudo-F | p | Unique perms |
|---|---|---|---|---|---|---|
| Categories (C) | 2 | 2.915 | 1.457 | 6.174 | **<0.001** | 9938 |
| Time (T) | 1 | 5.993 | 5.983 | 25.389 | **<0.001** | 9937 |
| C x T | 2 | 2.665 | 1.332 | 5.645 | **<0.001** | 9952 |
| Residuals | 35 | 8.261 | 0.236 | | | |

**Table 2. Pairwise analysis on the two-way PERMANOVA testing for the holobiont physiological traits between coral categories (BZ, BB, RZ) and sampling times (T1 and T2).** Similarities percentage analysis (SIMPER) was used to identify which physiological traits explained the largest portion of the variance. P-values in bold are significant.

| PAIRWISE and SIMPER | | | | | |
|---|---|---|---|---|---|
| | Category | T | P | Average Diss. | % Contribution |
| **T1 *vs* T2** | BZ | 3.660 | **0.001** | | |
| | BB | 2.943 | **0.007** | | |
| | RZ | 2.879 | **<0.001** | | |
| **Category x T1** (Bleaching) | | | | | |
| | BB *vs* BZ | 3.447 | **<0.001** | 34.3% | 18.3% Symbio. |
| | | | | | 16.78% Pg:R |
| | | | | | 11.2% rETR$_{max}$ |
| | BB *vs* RZ | 2.79 | **<0.001** | 33.3% | 19.3% Pg:R |
| | | | | | 16.2% Symbio. |
| | | | | | 10.5% rETR$_{max}$ |
| | BZ *vs* RZ | 1.916 | **0.009** | 13.6% | 18.6% Carbo. |
| | | | | | 15.3% Yield$_{max}$ |
| | | | | | 12.8% Lipids |
| **Category x T2** (Post-bleaching) | | | | | |
| | BB *vs* BZ | 0.891 | 0.584 | 11.0% | |
| | BB *vs* RZ | 1.775 | **0.011** | 13.6% | 20.3% Carbo. |
| | | | | | 12.6% Biomass |
| | | | | | 10.8% Proteins |
| | BZ *vs* RZ | 3.153 | **<0.001** | 15.0% | 22.5% Carbo. |
| | | | | | 13.3% Biomass |
| | | | | | 12.3% R |

and biomass (Fig 3D, 3I, 3L and 3M; Table 3). One-way PERMANOVA was used to test if the colonies from BB and BZ that survived at T2 recovered their reserves between T1 and T2 (S3 Table; Fig 4). Data underlined the capacity of both BB and BZ to increase their biomass and protein over a four-month time window (Fig 4A and 4C), likely depleting their energy reserves since both carbohydrates and lipids were lower at T2 (Fig 4B and 4D). The RZ colonies significantly decreased their lipids and carbohydrates from T1 to T2 (Fig 4B and 4D), although there were not changes in protein and biomass (Fig 4A and 4C).

## Effect of stress on the Symbiodiniaceae community

Two-way PERMANOVA showed significant differences in the Symbiodiniaceae ITS2 type profiles and major sequences between sampling times, coral categories, and their interaction (Table 4; S3 and S5 Figs).

The nMDS showed clear separation between times in BB colonies compared to the other two categories (BZ and RZ) (S5 Fig), which was confirmed by the pairwise analysis. During the bleaching event (T1), all of the bleached colonies had 93.8% difference in their ITS2 sequences compared to the unbleached colonies in Bouraké (Table 5). SIMPER analysis suggested that the major differences in ITS2 sequences between times were due to the presence of C50b, C50p and C3, with the major profile C50b.C50p.C3.C3bm.C50f in the BB category (Table 5; Fig 5).

Notably one BB coral had a major type profile of C3k.C3bo.C50a.C3ba.C50q. In contrast, the BZ category was represented mostly by C1, C1b and C1c, with the major ITS2 type profile C1.C1b.C1c.C42.2.C1bh.C1br.C1cb.C3. We did not detect any of the BZ major sequences in

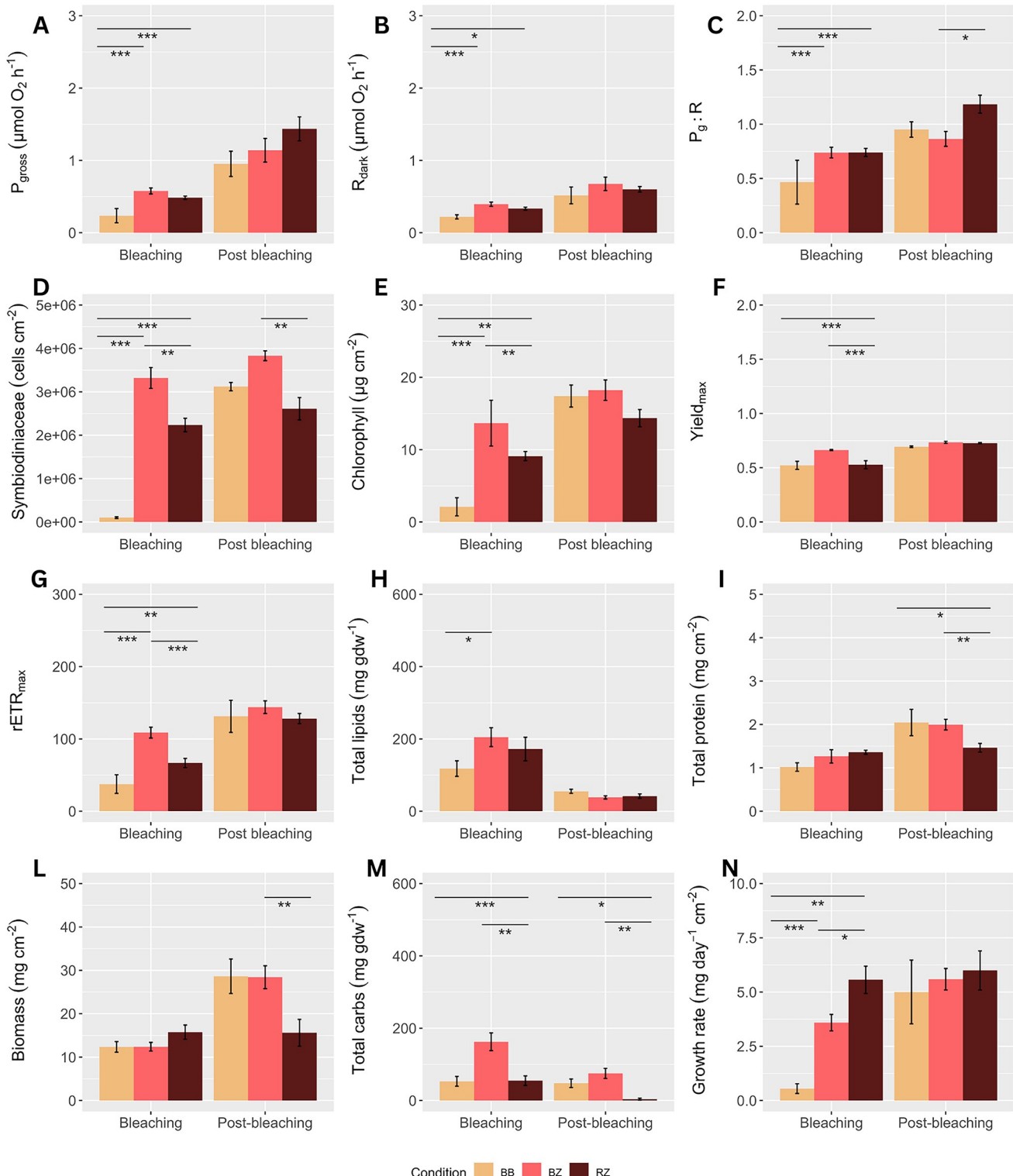

**Fig 3. Holobiont physiological traits.** Physiological traits (mean ±SD) in BB (Yellow, Bouraké bleached), BZ (Red, Bouraké healthy), and RZ (Brown, reference healthy) colonies of *Acropora muricata*, during bleaching (T1) and post bleaching (T2). Asterisks above barplots indicate the significance of post-hoc tests when one-way PERMANOVA was significant (* = 0.01, ** = 0.001, *** = < 0.001). At T1, the number of replicates were n = 15 for BB and BZ and n = 9 for RZ, while at T2 replicates were n = 3, n = 12, and n = 6 for BB, BZ, and RZ, respectively.

**Table 3. One-way PERMANOVA between coral categories (BB, BZ, RZ) for each coral phenotypic trait at T1 (bleaching), and T2 (post-bleaching).** P-values in bold are significant.

| Time T1 (bleaching) | | | | | | | |
|---|---|---|---|---|---|---|---|
| Trait | df | SS | MS | Pseudo-F | $p$ | | Permutations |
| $P_{gross}$ | (2,36) | 1.793 | 0.896 | 54.75 | < **0.001** | | 9946 |
| $R_{dark}$ | (2,36) | 0.233 | 0.116 | 9.98 | <**0.001** | | 9955 |
| $P_{gross}$:R | (2.36) | 1.670 | 0.835 | 27.83 | <**0.001** | | 9947 |
| Chl | (2,18) | 892.6 | 444.320 | 21.81 | <**0.001** | | 9952 |
| Symbio. | (2,18) | 8.791 | 4.395 | 227.3 | <**0.001** | | 9955 |
| $Yield_{max}$ | (2,32) | 0.076 | 0.038 | 9.15 | **0.001** | | 9957 |
| $rETR_{max}$ | (2,32) | 158.82 | 79.4 | 17.49 | <**0.001** | | 9937 |
| Biomass | (2,36) | 1.429 | 0.714 | 2.22 | 0.126 | | 9954 |
| Lipids | (2,36) | 0.102 | 0.051 | 3.63 | **0.036** | | 9951 |
| Proteins | (2,18) | 0.096 | 0.048 | 2.96 | 0.075 | | 9951 |
| Carbo. | (2,36) | 0.281 | 0.14 | 10.90 | <**0.001** | | 9946 |
| Calcif. | (2,21) | 6.178 | 3.089 | 21.57 | <**0.001** | | 9960 |
| **T2 (Post-bleaching)** | | | | | | | |
| $P_{gross}$ | (2,18) | | 0.125 | 0.062 | 1.37 | 0.279 | 9945 |
| $R_{dark}$ | (2,18) | | 0.022 | 0.011 | 0.49 | 0.635 | 9940 |
| $P_{gross}$:R | (2,18) | | 0.106 | 0.053 | 4.43 | **0.025** | 9951 |
| Chl | (2,18) | | 1.1E6 | 5.9E6 | 1.63 | 0.245 | 9943 |
| Symbio. | (2,18) | | 7.290 | 3.645 | 8.47 | **0.002** | 9948 |
| $Yield_{max}$ | (2,18) | | 0.001 | 0.006 | 2.96 | 0.082 | 9939 |
| $rETR_{max}$ | (2,18) | | 1.883 | 0.941 | 0.57 | 0.546 | 9963 |
| Biomass | (2,18) | | 8.887 | 4.443 | 3.97 | **0.039** | 9960 |
| Lipids | (2,18) | | 0.043 | 0.021 | 1.50 | 0.258 | 9929 |
| Proteins | (2,18) | | 0.177 | 0.088 | 4.60 | **0.022** | 9951 |
| Carbo. | (2,18) | | 0.171 | 0.085 | 11.16 | **0.001** | 9927 |
| Calcif. | (2,18) | | 2.009 | 1.004 | 0.26 | 0.784 | 9960 |

13 out of 15 BB colonies during the bleaching. However, one BB colony that survived the stress event had a small percentage of C1, C1b and C1c in their ITS2 major sequences at T1 (Fig 5). Two BZ colonies had a little percentage of C50b and C3bm, but their ITS2 major sequences were still dominated by C1, C1b and C1c at T1, which were absent from the major sequences at T2. The two unbleached categories (BZ and RZ) were 95.2% different from each other (Table 5). Indeed, ITS2 sequences and profiles of RZ were more diverse compared to the unbleached colonies of the Bouraké lagoon, with colonies represented by C50b, C3k, and C21 as the main ITS2 sequences in their community (Fig 5). After the bleaching (T2) the three BB colonies showed the same Symbiodiniaceae community of BZ colonies, therefore dominated by C1, C1b and C1c as major ITS2 sequences, and with the major ITS2 type profile C1.C1b. C1c.C42.2.C1bh.C1br.C1cb.C3. Notably, three BZ colonies had different ITS2 type profiles from T1 to T2 (Table 5; Fig 5).

## Discussion

During January 2021, a coral bleaching event followed by moderate mortality of the family Acroporidae was observed in Bouraké but not at other localities in New Caledonia. This event occurred during a particularly warm and rainy period associated with La Niña. As a result of the topography of the Bouraké semi-enclosed lagoon, temperatures reached the extreme values

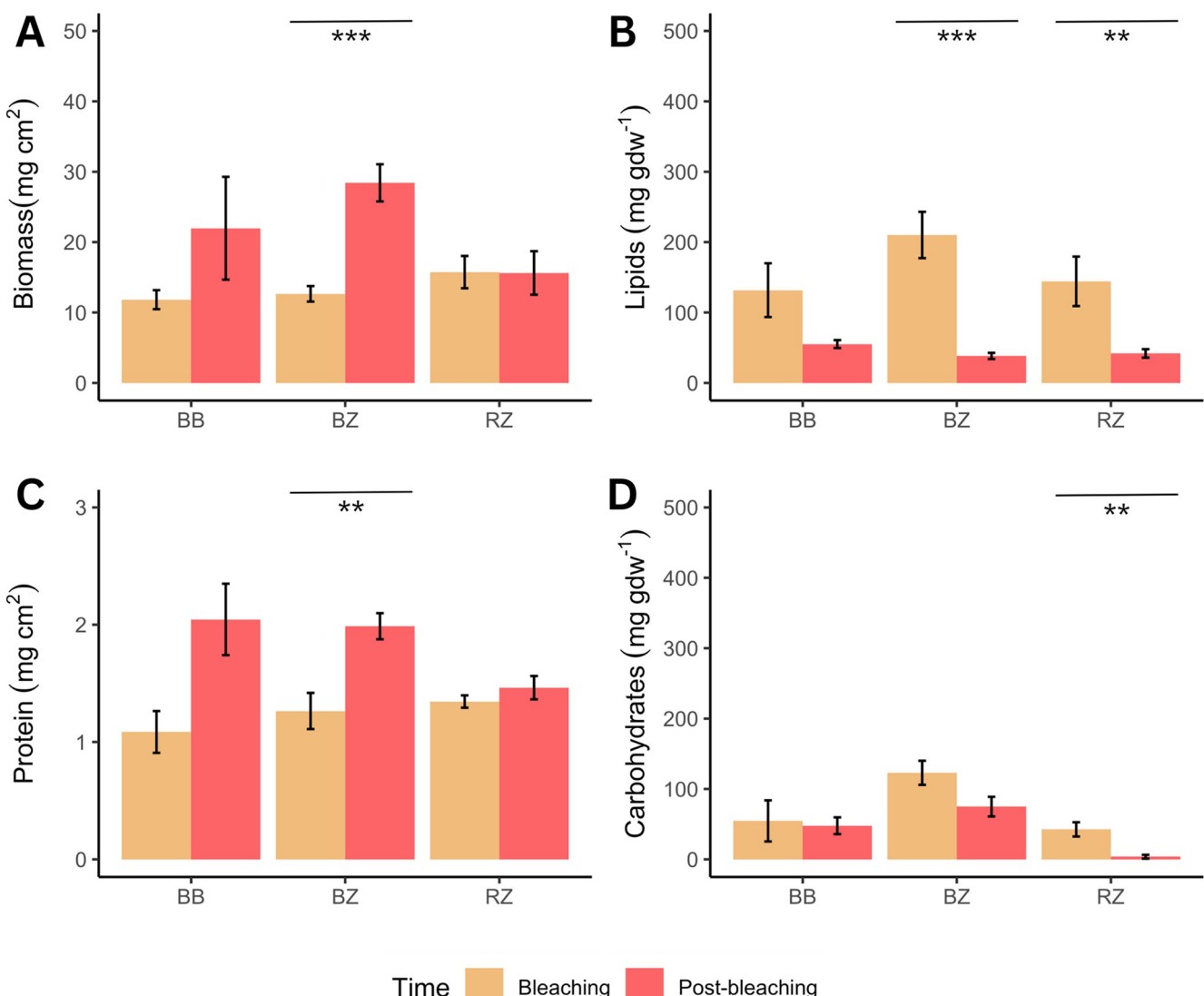

**Fig 4. Trend in energetic reserves of coral over the two timepoints.** Changes in biomass and energetic reserves (mean ±SD) in BB (Bouraké bleached), BZ (Bouraké healthy), and RZ (reference healthy) colonies of *A. muricata* during bleaching (yellow) and post-bleaching (orange). Asterisks above barplots indicate the results of post-hoc tests when one-way PERMANOVA was significant (* = 0.01, ** = 0.001, *** = < 0.001). Only the colonies that survived/found were considered (n.3, 12, and 6 for BB, BZ and RZ).

of 32.89˚C, nearly two degrees higher than the reference reef (T = 30.95˚C), which in combination with heavy rain likely caused the observed bleaching and coral mortality. This observation was unexpected since corals inhabiting the semi-enclosed lagoon of Bouraké have been described to cope with chronic exposures to low seawater pH, deoxygenation, and warm temperatures [33]. Unfortunately, we were unable to disentangle the main causes of the bleaching, i.e., seawater warming vs freshwater input. A mangrove lagoon on the Great Barrier housing extreme corals recently experienced loss of *Pocillopora acuta* species due to freshwater input from heavy rain, highlighting that extreme coral systems are susceptible to periodic stress events [39]. In Bouraké, regardless of the cause of coral bleaching, it was evident that despite the more extreme environmental conditions, unbleached colonies had higher photo-physiological performances than colonies from the reference reef and a marked difference in their Symbiodiniaceae communities. In contrast, colonies that bleached in Bouraké, had a similar

**Table 4. Two-way PERMANOVA on the relative abundance of ITS2 sequence types, and major ITS2 profiles of *Acropora muricata* among categories (BB, BZ, and RZ) and between times (T1 and T2).** P-values in bold are significant.

| Trait | df | SS | MS | Pseudo-F | *p* | Permutations |
|---|---|---|---|---|---|---|
| **ITS2 abundances** | | | | | | |
| Time (T) | 1 | 1.7908 | 1.790 | 14.532 | **<0.001** | 9944 |
| Categories (C) | 2 | 13.942 | 6.971 | 56.568 | **<0.001** | 9950 |
| T x C | 2 | 4.2088 | 2.104 | 17.077 | **<0.001** | 9956 |
| Residuals | 54 | 6.6545 | 0.123 | | | |
| **ITS2 profiles** | | | | | | |
| Time (T) | 1 | 11145 | 11457 | 5.4092 | **0.006** | 9950 |
| Categories (C) | 2 | 76424 | 38212 | 18.041 | **<0.001** | 9947 |
| T x C | 2 | 27745 | 13873 | 6.5996 | **<0.001** | 9942 |
| Residuals | 54 | 11226 | 2118 | | | |

symbiont community to the colonies at the reference site. Interestingly, the Bouraké colonies that recovered from the stress event returned their normal physiological performances, and the Symbiodiniaceae communities were similar to the unbleached colonies. Recovery post bleaching was likely promoted by energy reserve catabolism for both bleached and unbleached corals [69,70].

**Table 5. Pairwise analysis on the two-way PERMANOVA testing for the ITS2 sequences and profile among coral categories (BZ, BB, RZ) and sampling times (T1 and T2).** Similarities percentage analysis (SIMPER) was used to identify which sequences explained the largest portion of the variance. P-values in bold are significant.

| PAIRWISE and SIMPER | | Category | T | P | Average Diss. | % Contribution |
|---|---|---|---|---|---|---|
| **T1 *vs* T2** | | BZ | 0. 9082 | 0.489 | | |
| | | RZ | 1.0058 | 0.391 | | |
| | | BB | 16.52 | **0.001** | | 34.5% C50b |
| | | | | | | 8.6% C3 |
| | | | | | | 7.7% C50p |
| **Category *T1 (Bleaching)** | | | | | | |
| | | BB *vs* BZ | 27.559 | **<0.001** | 93.8% | 38.5% C50b |
| | | | | | | 31.2% C1b |
| | | | | | | 31.1% C1 |
| | | BB *vs* RZ | 6.4014 | **0.001** | 95.1% | 37.4% C50b |
| | | | | | | 16.2% C21 |
| | | | | | | 12.0% Others |
| | | | | | | 10.6% C3k |
| | | BZ *vs* RZ | 2.9248 | **<0.001** | 48.1% | 31.1% C1 |
| | | | | | | 20.6% C50b |
| | | | | | | 8.0% C21 |
| **Category *T2 (Post-bleaching)** | | | | | | |
| | | BB *vs* BZ | 0.8213 | 0.471 | | |
| | | BB *vs* RZ | 4.1733 | **0.013** | 95.7% | 31.9% C50b |
| | | | | | | 30.5% C1 |
| | | | | | | 5.7%C1b |
| | | BZ *vs* RZ | 8.388 | **<0.001** | 98.8% | 30% C1 |
| | | | | | | 31.9% C50b |
| | | | | | | 4.9% C1b |

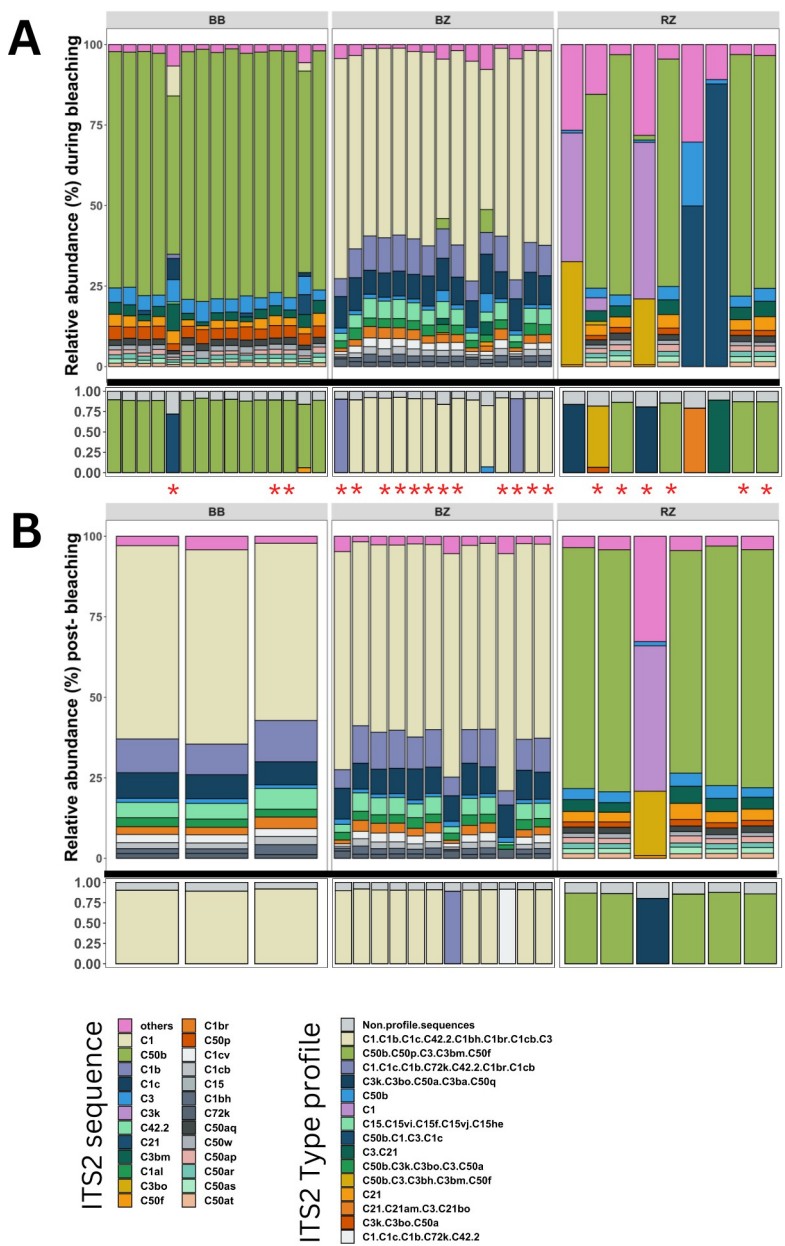

**Fig 5. Recovered ITS2 sequences and predicted ITS2 type profiles.** A) ITS2 sequence relative abundance (%), and predicted major ITS2 type profiles of *Acropora muricata* for BB (Bouraké bleached), BZ (Bouraké healthy), and RZ (reference healthy) during bleaching (T1), and B) post bleaching in (T2). Results are plotted as stacked bar charts with a single column representing a sample. For each column in the stacked bar plots, the relative abundances in percentage of ITS2 sequences are plotted above the horizontal black line, whilst the predicted ITS2 profiles are plotted below. Only the most 25 major ITS2 sequences are shown, the remaining sequences are represented by "others" (pink portion of the stacked bar chart). Red asterisks indicate the colonies that were found alive at T2. Recognizable sequences (e.g., C1, C50b or C3) refer to sequences that have been used to characterize ITS2 type profiles as a part of previous analyses that have been run through the SymPortal analytical framework.

## Symbiont community as indicator of resistance to bleaching

Although the three coral categories (BB, BZ, and RZ) were all dominated by *Cladocopium* spp., we found significant differences in the main Symbiodiniaceae ITS2 type profiles and sequences between sites and among categories during the bleaching (T1). In contrast, after bleaching

(T2), we only observed significant differences between sites. Indeed, the major Symbiodiniaceae ITS2 type sequences of bleached corals (BB), which were 93.82% different from unbleached colonies in Bouraké (BZ), were C50b, C3, and C50p (type profile: C50b.C50p.C3. C3bm.C50f). Interestingly, the three bleached colonies that recovered acquired the same ITS2 type profiles as the nonbleached (BZ) colonies. These results suggest two main findings: 1) *Cladocopium proliferum* (formerly *C. goreaui*/ITS2 type C1 or C1[acro]; [65,71]) appears to facilitate resistance to bleaching in "extreme" corals; and 2) "extreme" corals appear capable of altering their symbiont community following bleaching.

*Cladocopium* is one of the most diverse genera in the Symbiodiniaceae [72] with functional variation in symbiont thermal performances [73], and functional variation in gene expression across reefs [74,75]. *C. proliferum* has already been described as a dominant symbiont species in healthy colonies of *A. muricata* from Bouraké [76]. Camp et al. [76] collected coral samples housing *C. proliferum* in 2016, during the first documented heat wave reported in New Caledonia that caused widespread coral bleaching. It is therefore likely that corals in Bouraké, and their *C. proliferum*, have previously been exposed to severe thermal stress. Several heat-evolved (i.e. thermally selected) strains of *C. proliferum* were found to display faster growth rates and higher photosynthetic efficiency than their wild-type counterparts under elevated temperatures [77,78]. Similarly, some heat-evolved *C. proliferum* strains also enhanced bleaching tolerance when in symbiosis with coral larvae and recruits [79,80]. Previous studies [18,77] have also demonstrated the improved thermo-tolerance and physiological performances of corals hosting *C. proliferum* in controlled conditions by testing the effect of temperature only. Our data provides new ecologically relevant evidence showing that corals already hosting thermo-tolerant *C. proliferum* are able to better face extreme conditions, and maintain host-related physiological traits, to the combined effects of high temperature, low pH, low oxygen, and variable salinity.

The second major finding is represented by the change in the symbiont community of the bleached colonies that survived post-bleaching. This change may be due either to the switching from the original to a new, de novo symbionts uptake from the environment, and more resistant symbiont community [81], or to the shuffling of cryptic symbiont types in the existing symbiont communities proliferating post-bleaching [82,83]. In our study, *C. proliferum* was observed during the bleaching as a very minor species in only one of the three BB colonies that survived, suggesting a switching toward the opportunistic *C. proliferum*. However, the limited sample size, together with the few examples existing in the literature [84,85], does not allow us to conclude which strategy *A. muricata* used. Whatever the mechanism, it was suggested that bleaching can provide an adaptive ecological opportunity for corals by allowing the proliferation of more resilient symbiont species [86]. This strategy could help *A. muricata* to survive stress events, but on the other hand the reduction in the Symbiodiniaceae diversity after recovery, could also be a disadvantage in a multi-stress environment such as Bouraké.

## Coral host physiology during and after bleaching

We found that colonies of the same species, living in the same environment in Bouraké, had a different physiological response and suffered differently during the stress period. Only three bleached BB colonies survived the stress event, while twelve out of the fifteen unbleached colonies (BZ) survived. In contrast, colonies at the reference site (R1) did not show any sign of visual bleaching during the period of stress, nor did they undergo partial mortality during the post bleaching period.

One of the main physiological expenditures affected during a period of stress, and which defines a coral's health status is its somatic growth. Growth rate is tightly linked with

photosynthesis (i.e., light-enhanced calcification; [87]), and to the efficiency of certain Symbiodiniaceae to translocate photosynthetic compounds to their hosts [88]. When a coral undergoes thermal stress, photosynthesis is first halted, and then Symbiodiniaceae are expelled, depriving the host of essential photosynthetic carbon [9]. Accordingly, we found that during T1, growth rates was slightly reduced for unbleached colonies in Bouraké, and dramatically compromised for bleached colonies compared to reference colonies. Therefore, both unbleached and bleached colonies in Bouraké underwent stress but had different plasticity in their physiological responses. We did not expect that growth rate in *A. muricata* was lower in Bouraké than in the reference colonies since, in a recent study, Bouraké colonies had greater calcification compared to reference corals at seawater pH ranging from 7.5 to 8.1 [45]. It is important to note that in the present study not only were corals from Bouraké chronically exposed to extreme and fluctuating seawater pH, but they also had to withstand the combination of acidification with acute and prolonged thermal stress (and likely an osmotic shock due to a decrease drop in the seawater salinity). Our findings agree with [89], who found that while growth rates of tropical coral species increased under high $p$CO$_2$ conditions, they decrease drastically when corals are exposed to the combined effects of high $p$CO$_2$ and high temperatures (31°C), which caused the loss of their symbionts. Similarly, another study reported 50% reduction in calcification when *Stilopora pistillata* was exposed to the combined effects of temperature and decreased pH [90].

In our study, all the photo-physiological traits (i.e., rETRmax, Yeld$_{max}$, Pg:R, Symbiodiniaceae and chlorophyll contents) were highly compromised in the bleached category (BB) at the time of the bleaching (T1). This is not surprising since prolonged exposure to a period of stress can cause a dysfunction between symbionts and host [91] resulting in the loss of symbionts, i.e., bleaching [92–95], and the consequent loss of photosynthetic carbon available to the host [9]. Indeed, photosynthesis and respiration rates of bleached colonies were significantly compromised, suggesting that the carbon used for respiration in bleached colonies was acquired through the catabolism of energetic reserves and/or heterotrophy [96], while metabolic activities and notably dark respiration, were lowered likely to decrease the risk of mortality [17]. However, 80% of the bleached colonies died, suggesting that 1) the stress was too intense and/or persistent for the coral to survive; 2) the catabolism of energetic reserves and/or heterotrophic ability were unable to counter the impacts of the stress event. Interestingly, unbleached colonies from Bouraké (BZ) showed the highest photo-physiological performances (i.e., Symbiodiniaceae and chlorophyll contents, Yield$_{max}$, and rETR$_{max}$) when compared to the reference site (RZ), despite the higher environmental stress in Bouraké. The high photo-physiological performances of BZ may be due to the characteristic of *C. proliferum*, making the holobiont more thermo-resistant and improving its basal physiology [78]. For instance *Cladocopium* sub-type C1 is quite efficient in fixing photosynthetic carbon and increasing nitrogen acquisition [97], which can boost the photo-physiological traits of the Symbiodiniaceae community [98–100].

The three bleached colonies (BB) that survived post-bleaching fully recovered from the stress. Indeed, at T2, they had a density of Symbiodiniaceae, chlorophyll, and photosynthetic efficiency of the photosystem II (both ETR$_{max}$ and Yield$_{max}$) comparable to unbleached colonies in Bouraké (BZ). The new Symbiodiniaceae acquired may have helped the host to recover normal physiological performances. BZ colonies retained a significantly higher Symbiodinaceae density than the reference colonies (RZ), while all of the other traits including growth rate were still comparable after bleaching. The high density of Symbiodiniaceae could have induced self-shading and thus reduced the light availability [101] for photosynthesis. This might explain why the P$_{gross}$ was not higher in unbleached and ex-bleached colonies of Bouraké when compared to reference colonies. This result is consistent with the photosynthetic rates measured on many coral species in Bouraké [35,45].

In terms of energetic reserves, which have been demonstrated to promote coral recovery from bleaching [16,17,70,96], we found that both lipids and carbohydrates in bleached colonies (BB) were significantly lower than the unbleached colonies in Bouraké during the stress event (T1). The finding suggests that these two were the main source of energy for bleached colonies. At the same time, unbleached colonies from Bouraké (BZ) had similar lipid content and distinctly higher carbohydrates than the reference colonies. Carbohydrates are typically acquired by autotrophy and are rapidly depleted [102,103], while lipids are longer-lasting energy reserves [14,96] that can be used to produce energetic metabolites such as ATP [104]. Carbohydrates are one of the first products of photosynthesis, and the greater concentration in BZ during the bleaching could be linked to the improved photo-physiological traits compared to the reference colonies. In agreement with previous studies [31,70,96,105], both biomass and proteins did not differ among categories of colonies during the bleaching, but they increased in Bouraké colonies after the bleaching period. Biomass is usually an indicator of coral health [96], while proteins are involved in enzymatic reactions and biomineralization of the skeleton [106], and their increase after bleaching can provide evidence of recovery from a stress event [52,107].

## Conclusions

This study gives more insights into the mechanisms used by the widespread coral *A. muricata* to cope with and evolve in an extreme environment during an acute stress event. Indeed, while corals from Bouraké have been suggested to be resilient to acidification, deoxygenation, and warming, it is unknown if the acquired resilience will help them cope with further stress. This study provides ecological observations that complement the existing literature about the role of stress-tolerant symbionts in corals that already live at the edge of their perceived environmental limits. Our data also confirm the role of energy reserves during bleaching in "extreme" corals. First, corals use carbohydrates to cope with the stress and then lipids to compensate for the lack of autotrophic carbon. Overall, our findings sustain the hypothesis that a specific Symbiodiniaceae community can support long-term acclimatization allowing corals to persist in such harsh conditions. Natural laboratories where coral communities live under extreme conditions are becoming crucial tools because they provide precious insights on the tolerance of symbionts, corals, and host-symbiont associations. These locations also tell us which species are likely to survive under the combined effect of multiple environmental drivers (i.e., fluctuating pH, deoxygenation, and osmotic shock during a heat stress event), suggesting which species will likely persist in the future.

## Supporting information

**S1 Table. Mortality (%) of tagged colonies of *Acropora muricata*.** Bouraké healthy (BZ), Bouraké bleached (BB), and reference healthy (RZ) in May 2021 (T2) according to their initial category and origin.
(DOCX)

**S2 Table. Pairwise comparison on coral physiological traits.** T-test on one-way PERMANOVA for each coral physiological trait measured in *Acropora muricata* during the bleaching (T1) and post-bleaching (T2).
(DOCX)

**S3 Table. Pairwise comparison on metabolic reserves.** T-test on one-way PERMANOVA for each metabolic reserve (i.e., proteins, lipids, carbohydrates, and biomass) measured in

*Acropora muricata* post-bleaching (T2). Only the colonies that survived at T2 were considered. (DOCX)

**S1 Fig. Study sites. A)** Location of the two study sites: the Bouraké semi-enclosed lagoon (B2) and the reference site (R1). **B)** Photographs of *Acropora* spp. assemblage during the bleaching event in January 2021 at Bouraké (site B2). Satellite data were downloaded from www.georep. nc (©Georep contributors) and customized in QGIS (version 3.4.14).
(TIFF)

**S2 Fig. Rain regime.** Daily rainfall regime in Bouraké region measured from the 1st of December 2020 to the 31st May 2021. Data were recorded from Meteo France in proximity of our study sites.
(TIFF)

**S3 Fig. ITS2 rarefaction curve.** Rarefaction curve on number of sequences and number of Derived Intragenomic sequences Variance of ITS2.
(TIFF)

**S4 Fig. Physiological profiles.** Non-parametric multidimensional scaling plot (nMDS) of holobiont physiological traits in *Acropora muricata* measured at T1 (circles) and T2 (triangles), corresponding to the three categories of colonies BB (Green, Bouraké bleached colonies), BZ (Blue, Bouraké healthy colonies), and RZ (Black color, reference healthy colonies).
(TIFF)

**S5 Fig. ITS2 relative abundance.** Non-parmetric multidimensional scaling plot (NMDS) of the coral relative abundance of ITS2 sequence types during T1 (bleaching, circles) and T2 (post bleaching, triangles). Green corresponds to BB (Bouraké bleached colonies), blue to ZB (Bouraké healthy colonies), and black to RZ (reference R1 healthy colonies).
(TIFF)

## Acknowledgments

We are grateful to Clément Tanvet, Robin Quéré, Celia Lemeu, Florence Antypas, the staff of the research station IFREMER- St. Vincent, and the staff of the Plateforme du vivant (UMR-Entropié) for hosting us during sample processing. We thank Jordi Giraud and Mahe Dumas for their help during the fieldwork, as well as Greg and Esmé for the accommodation in Bouraké. We also thank the four reviewers that helped to improve the quality of this manuscript. We are indebted to the Province Sud for coral collection permits (SuperNatural, agreement #3413–2019).

## Author Contributions

**Conceptualization:** Cinzia Alessi, Riccardo Rodolfo Metalpa.

**Data curation:** Cinzia Alessi.

**Formal analysis:** Cinzia Alessi, Nelly Wabete.

**Funding acquisition:** Emma F. Camp, Riccardo Rodolfo Metalpa.

**Investigation:** Cinzia Alessi, Emma F. Camp, Claude Payri.

**Methodology:** Cinzia Alessi, Nelly Wabete, Riccardo Rodolfo Metalpa.

**Supervision:** Hugues Lemonnier, Riccardo Rodolfo Metalpa.

**Visualization:** Cinzia Alessi.

**Writing – original draft:** Cinzia Alessi, Riccardo Rodolfo Metalpa.

**Writing – review & editing:** Hugues Lemonnier, Emma F. Camp, Nelly Wabete, Claude Payri.

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
