## [Decision Letter · Decision Letter 0]

31 Aug 2023

PONE-D-23-04677Intra-clade diversity provides resistance to coral bleaching under extreme environmental conditions in Acropora muricataPLOS ONE

Dear Dr. alessi,

Thank you for submitting your manuscript to PLOS ONE. After careful consideration, we feel that it has merit but does not fully meet PLOS ONE’s publication criteria as it currently stands. Therefore, we invite you to submit a revised version of the manuscript that addresses the points raised during the review process.

ACADEMIC EDITOR:Hello,

   Sorry that this has taken so long, but it was passed off to me from another editor, and that editor had only had it reviewed by your collaborators! I left their comments (reviewers #1 and 2), but of course they are more minor. Therefore, I would suggest to focus more on reviewers #3-4, who are experts in this field. 

Anderson

International Coral Reef Society

Coral Research and Development Accelerator Platform

Coral Reef Diagnostics

We look forward to receiving your revised manuscript.

Kind regards,

Anderson B. Mayfield, Ph.D.

Academic Editor

PLOS ONE

“We are grateful to Robin Quéré, Celia Lemeu, Florence Antypas and the staff of the research station IFREMER- St. Vincent for hosting us during the sampling. We thank Jordi Giraud and Mahe Dumas for their help during the fieldwork, as well as Greg and Esmé for the accommodation in Bouraké. We are indebted to the Province Sud for coral collection permits (SuperNatural, agreement #3413-2019). CA was supported by a PhD fellowship from Labex CORAIL and IFREMER. The research was supported by the French Ministry of Foreign Affairs, Fonds Pacifique project “SuperCoraux” #6614A1. Sequencing costs and the contribution of EFC to the project were supported by a CPDRF grant awarded to EFC.”

“CA was supported by a PhD fellowship from Labex CORAIL and IFREMER. The research was supported by the French Ministry of Foreign Affairs, Fonds Pacifique project “SuperCoraux” #6614A1. Sequencing costs and the contribution of EFC to the project were supported by a CPDRF grant awarded to EFC. Include this sentence at the end of your statement: The funders had no role in study design, data collection and analysis, decision to publish, or preparation of the manuscript.”

6. Please include a separate caption for each figure in your manuscript.

7. We note that Figure S1 in your submission contain [map/satellite] images which may be copyrighted. All PLOS content is published under the Creative Commons Attribution License (CC BY 4.0), which means that the manuscript, images, and Supporting Information files will be freely available online, and any third party is permitted to access, download, copy, distribute, and use these materials in any way, even commercially, with proper attribution. For these reasons, we cannot publish previously copyrighted maps or satellite images created using proprietary data, such as Google software (Google Maps, Street View, and Earth). For more information, see our copyright guidelines: http://journals.plos.org/plosone/s/licenses-and-copyright.

a. You may seek permission from the original copyright holder of Figure S1 to publish the content specifically under the CC BY 4.0 license. 

8. Please include your tables as part of your main manuscript and remove the individual files. Please note that supplementary tables (should remain/ be uploaded) as separate "supporting information" files

Additional Editor Comments:

Hello,

Sorry that this has taken so long, but it was passed off to me from another editor, and that editor had only had it reviewed by your collaborators! I left their comments (reviewers #1 and 2), but of course they are more minor. Therefore, I would suggest to focus more on reviewers #3-4, who are experts in this field.

Anderson

International Coral Reef Society

Coral Research and Development Accelerator Platform

Coral Reef Diagnostics

Reviewers' comments:

Reviewer's Responses to Questions

**Comments to the Author**

1. Is the manuscript technically sound, and do the data support the conclusions?

Reviewer #1: Yes

Reviewer #2: Yes

Reviewer #3: Partly

Reviewer #4: Partly

2. Has the statistical analysis been performed appropriately and rigorously? 

Reviewer #1: Yes

Reviewer #2: Yes

Reviewer #3: Yes

Reviewer #4: Yes

3. Have the authors made all data underlying the findings in their manuscript fully available?

Reviewer #1: Yes

Reviewer #2: No

Reviewer #3: Yes

Reviewer #4: No

4. Is the manuscript presented in an intelligible fashion and written in standard English?

Reviewer #1: Yes

Reviewer #2: Yes

Reviewer #3: Yes

Reviewer #4: Yes

5. Review Comments to the Author

Reviewer #1: This study helps to improve our knowledge on the disparity of coral colony responses to marine heat waves, why within the same species, certain colonies resist better than others to heat stress? This manuscript provides crucial information allowing us to improve the forecasts about the future of coral reefs but also new useful knowledge to try to establish lines of resistant corals. In this manuscript, the results showing that the few bleached colonies that survived acquired the same ITS2 profiles of the unbleached resistant colonies are extremely interesting, they confirm the essential role of Symbiodiniaceae in resistance/resilience to heat stress and these results are essential to guide future protective measures. I also appreciated the honesty of the authors, who recognize that they cannot decipher if the bleaching was due to seawater warming or freshwater input and do not rule out any possibility.

The conclusions of this manuscript are important and innovative so I recommend it for publication, I only have minor corrections/suggestions listed below

Reviewer #2: Review

In the article entitled “Intra-clade diversity provides resistance to coral bleaching under extreme environmental conditions in Acropora muricata” Alessi et al present the results of an integrative study aiming at understanding potential mechanisms underlying the physiological plasticity of corals that live in extreme and fluctuating conditions during a period of acute stress. To address this question authors performed series of physiological characterization on the cnidarian host and Symbiodiniaceae symbiont on 15 bleached and 15 unbleached corals from an “extreme” site and 15 unbleached corals from a reference site. These data set were analyzed with appropriate bioinformatics and statistical methods and the results obtained are very interesting, welle presented and well discuss.

My conclusion is that this article deserves a publication in Plos One but see my minor comments below.

Minor comment:

- I think that the title should be moderate a little bit since a functional link between the symbiodiniaceae community composition and the phenotypic effect was not demonstrated. The term associated rather than provides would be used.

- Line 47 the sentence is a little bit confusing. Do it mean that the symbiont translocate up to 90% of its photosynthate, or that the symbiont cover up to 90% of the daily carbon needed or that because of the drastic reduction in zoox density the quantity of photosynthate carbon translocated decrease up to 90%?

- Line 249 no segments were underlined on the primers presented line 246-248

- Line 256 precise sequencing depth

- Line 372 It is important to provide basic sequencing metrics to show that the amount of sequence is enough to support the results. You can either provide a rarefaction curve in suppfile or simply a number of sequence. The OTU table or equivalent is also classically provided for metabarcoding experiments. Finally you must deposit your sequence data (FastQ files) on a public database (SRA for example).

- I don’t understand how you can link the survival of the 3 BB colonies with the change in ITS2 profile type. What is the cause what is the consequence? Do the stress have induce the change and will make the surviving BB colonies “BZ” colonies the next time, or is the change prior to the survivor to the stress event studied? For me it is quite difficult to conclude but; 1 the difference between the ITS2 profile type of the BZ and RZ at T1 and 2 the similarities between the ITS2 profile type of the BB and RZ at T1 is enough to support your conclusion. The similarities between BB and BZ at T2 is in my opinion the results of a post stress colonization enabled by free space as it was previously shown in other coral species ( for example https://doi.org/10.1111/gcb.12706)

- Line 562 I don’t understand how it can support adaptation (sensus heritable phenotypic trait). Do you suggest that there is a heritable basis behind the ability to change or not the symbiont community or the heterotrophic capability?

Congratulation for this very interesting work

Jeremie Vidal-Dupiol

note: I have recently decided to sign all my reviews because I believe the reviewers should be responsible for their review just as authors are responsible for their article.

Reviewer #3: This manuscript focuses on the thermal resilience of corals in New Caledonia and the links to symbiont community composition. The manuscript reports some interesting observations that suggest that a shift in the symbiont community composition, combined with internal lipid reserves, may facilitate survival under increased seawater temperatures, adding to an existing wealth of literature on this topic. Ultimately, I think that the manuscript will be acceptable, but I do have several major comments:

1) The manuscript contains a very large number of grammatical errors. I appreciate that the primary authors are not native English speakers, but I do note that one of the authors is - can you please make use of this fact and correct the English. I haven't listed all of the errors here (only some are included in my list below) as to do so would take far too long.

2) It's nice to see the authors use the old CZAR model, however they have not used it correctly. The crucial distinction between the CZAR and simply using a P:R ratio, is that the CZAR includes an estimation or measurement of photosynthate translocation from the symbionts to host. The authors don't use such a metric, so all they're really doing is presenting a daily P:R. Please adjust the manuscript accordingly.

3) This may or may not need changing, but the authors should double-check their Symbiodiniaceae nomenclature, and in particular how to refer to the various Cladocopium types. ITS2 sequence names can still be used where species names are not currently available, but there is convention around how these are stated - they might want to refer to this style guide: https://www.thelifeaquatic.net/?page_id=292

4) Of most concern is the impression given by the authors that symbiont 'switching' might be occurring in their study, yet no evidence is provided for this. It's much more likely that the shifts in symbiont community composition resulted from 'shuffling', with cryptic symbiont types in the existing symbiont communities proliferating post-bleaching, rather than de novo uptake of symbionts from the environment. That still might have happened, but there is so little evidence for this in the literature that great caution should be exercised when direct evidence cannot be provided.

Other comments

1) Line 57: The wording "by exchanging their microalgal symbiont" is too strong, as it implies symbiont switching (see my comment above)

2) Line 64: The holobiont also includes the microbiome, so adjust the definition here.

3) Line 71: "Acroporidae" shouldn't be italicized.

4) Line 74: Better as "when a mass bleaching event impacted entire..."

5) Line 79: Better as "The aim of this study was therefore..."

6) line 86: Symbiodiniaceae should start with a capital letter.

7) Line 100: Species composition of what exactly? Corals?

8) Line 104: What are the units for the DO values?

9) Line 120: Better as "...four main categories: healthy coral cover..."

10) Line 141 (and several other places): The correct word is "weighed" rather than "weighted"

11) Line 147: Why was coral mortality so high?

12) Line 149: Should be "to weigh fragments on a..."

13) Line 163: The specific PAM settings need to be stated here.

14) Line 205: What is meant by "sub-surface area"? This whole sentence is unclear.

15) Line 210 and elsewhere: All centrifuge speeds should be stated in g rather than rpm.

16) Line 222: Why were only 3 replicate cell counts measured? This is far too few - it's widely recognised that 8-10 replicate counts are needed for optimal accuracy.

17) Line 225: Should be "...10 ml of pure acetone WERE added".

18) Line 234: Better as "normalized to surface area."

19) Line 243: Unit needs to be superscripted.

20) Line 278 (and elsewhere): Should be "When significant differences..."

21) Line 289: Should this say "PERMANOVA and pairwise testing were..."?

22) Line 291: Should be "ANOVA was performed with....whereas PERMANOVA pairwise and..."

23) line 305: Should be "...since the year 1981."

24) Line 317: Better as "...no signs of polyp extroversion but the skeleton was still white and the coral not fully repopulated by algae"

25) Line 318: Should this say "old dead CORALS"?

26) Line 356: Better as "SOME of the initial colonies..."

27) Line 374 (and elsewhere0: Should be "THE data plot..."

28) Line 384 (and elsewhere): Should be "...more resilient TYPES". Be careful not to confuse the sequence nomenclature with the actual organism.

29) Line 388: Should be ""more DIVERSE compared..."

30) Line 399: Acroporidae shouldn't be italicized.

31) Line 423: Better as "which were 93.82% different..."

32) Line 427: Should be "which resulted in them being identical..."

33) lLine 437: Better as "IN the Symbiodiniaceae..."

34) Line 444: Should be "due TO the"

35) Line 448: Should be "reported IN corals"

36) Line 451: Should this say "Consistent with Davies et al."?

37) Line 453: Better as "have greater capacity to..."

38) Line 454: Should be "traitS"

39) Line 481: This sentence is a bit vague - in what way was calcification "affected"?

40) Line 487 (and elsewhere): either say "photosynthate" only or "photosynthetic carbon".

41) Line 495: Re-word as "ability was insufficient to cope with..."

42) Line 506: Should be "recovered FROM the stress"

43) Line 510: Better as "retained a significantly higher..."

44) Line 523: Should be "lipid content"

45) Line 529: What does "all more performant" mean?

46) Line 541: Better as "high load of organic carbon released..."

47) Line 552: Should be "...to cope WITH and evolve..."

48) Line 556: Should be "...UNDER the combined effect..."

49) Line 560: Should be "compensate FOR the lack..."

50) Lines 564-565: Here is another example of overstating the potential for symbiont switching rather than symbiont shuffling.

Reviewer #4: Alessi et al. observed a coral bleaching event taking place in Bourake lagoon in New Caledonia, an extreme reef exposed to highly variable environmental conditions. The sampled Acropora muricata colonies during and post-bleaching to monitor changes in symbiont community diversity, energetic stores, and various physiological parameters. The found that nearly all colonies that bleached shared the same heat-sensitive symbiont species, whereas nearly all colonies that did not bleach shared a different, putatively heat-tolerant symbiont species. Only three bleached colonies survived to the post-bleaching period, and all had transitioned to the alternate heat-tolerant symbiont species. Meanwhile, analyses of lipids, carbohydrates, lipids, proteins, biomass, photophysiology, symbiont density, chlorophyll, calcification, photosynthesis, and respiration all indicated distinct physiological characteristics for bleached vs. non-bleached vs. recovered vs. control colonies, as well as differences between colonies hosting different symbiont species. Based on these data, the authors conclude that variation in bleaching phenotypes is linked to symbiont community, which in turn influences recovery capacity among corals.

This experiment was competently performed. I don’t have any major concerns with the design or execution, although of course it is unfortunate that only three colonies survived in one of the treatments, which limits the scope of inference. The ecophysiological methods were all standard for coral studies and the statistics seemed appropriate. Overall, I feel the manuscript could be acceptable after a revision to address some of my concerns about the writing.

My issues all relate to the narrative. For one, there are dozens of studies like this with similar findings, where researchers observed a bleaching event and took samples to track coral physiology and symbiont dynamics during and post-bleaching, and found that different colonies responded differently, mediated in part by their symbiont communities. The authors, in my opinion, have not done a good job clarifying what makes this study unique, beyond the fact that it was a bleaching event in a particular lagoon in New Caledonia. Based on the title and introduction, I believe the important hook is that the Bourake site is considered an “extreme” reef, exposed to highly variable temperature, pH, dissolved oxygen, and salinity (which was probably more variable than usual thanks to intense La Nina rains during the study period). However, because this is just one extreme reef, it’s hard to draw conclusions about extreme reefs in general, rather than Bourake in particular. The authors don’t really tie their results back to the extreme nature of the reef, possibly because they didn’t measure many of the relevant variables directly, so this angle of the story loses focus. Instead, the manuscript starts to feel generic, often times simply reviewing what has been determined in other, similar studies, and noting that the results here are consistent. If the manuscript were a bit shorter and placed greater emphasis on how the extreme reef conditions may have shaped results, I think it would make a clearer novel contribution.

Moreover, because the reference site also experienced intense rainfall and the sample size for recovered colonies was only three, it’s hard to be confident that the results reported here are truly representative, even of Bourake. That would be ok if the story were framed in a narrow focus, but instead the authors seem to be drawing broad conclusions about different mechanisms of thermal tolerance, from the order in which different sources of energy are being used (even though there were only two time points), to the types of symbionts present (even though colonies were already bleaching when sampled and only three bleached colonies survived), to heterotrophy (not measured), to adaptation. Unfortunately, the authors were limited because they had to build a study around an ongoing bleaching event. Some of the questions could have been better addressed via more regular sampling or aquarium experiments. These limitations could have been better acknowledged, and some of the more speculative conclusions (e.g. about heterotrophy) could have been avoided. Of course, the flip side of the argument is that these are highly ecologically-relevant observations, and creating an extreme reef in an aquarium would be incredibly challenging.

Another major issue with the writing relates to how the Symbiodiniaceae are described. After the 2018 revision by LaJeunesse et al., the field has largely moved past referring to different symbiont groups as “clades,” which are now considered genera with their own names. The authors strangely combine both clade and genus terminology, at one point talking about “Symbiodinium Clade D” instead of “Durusdinium” but at another talking about “Cladocopium” instead of “Symbiodinium Clade C.” The title itself is misleading: when I read “intra-clade diversity provides resistance to coral bleaching” I honestly though the authors were talking about diversity within a coral clade. Instead, they really mean “intra-generic diversity among coral symbionts provides resistance to coral bleaching.”

From looking at the ITS2 profiles in Figure 5 it’s clear that there are only two main symbiont species in the BZ and BB corals (with a few rarer species in some colonies). I feel the discussion could be simplified if the authors referred to them as two species or, if they prefer, phylotypes. However, it’s wrong to speak as if each sequence variant (C1, C1b, C1c) is a separate entity (as is done in line 445, for example), given that the ITS2 array for a single species contains multiple intragenomic variants in specific proportions. A few other statements betray an older view of symbiont diversity. For example, L58 states that “some symbiont clades (i.e., Symbiodinium clade D) have been described as more resilient than others under elevated temperature.” But the whole group isn’t more heat tolerant, just some species within it. Better to say “some symbiont species (e.g. Durusdinium trenchii) are more resilient than others under elevated temperature.” Another example is L384: “suggesting an early uptake of more resilient sequences.” Corals don’t uptake sequences, they uptake symbionts. I know I’m being nitpicky, but these details are important to get right, otherwise confusion spreads. I’d highly recommend browsing the review by Davies et al. (2023) in PeerJ for a summary of the state of the art in our understanding of Symbiodiniaceae diversity.

Other minor concerns: I’d recommend adding + or – symbols in front of percentages when they are reported in the text. “Zooxanthellate” is a term for a species characteristic (e.g. reef-building corals are zooxanthellate while deep sea corals are usually azooxanthellate), so I find it strange to refer to non-bleached colonies as zooxanthellate (instead, perhaps call them “healthy” or “non-bleached” or “recovered” depending on the context). Some of the figures don’t really need color and might be clearer if depicted in grayscale. Certain elements of the methods could use more introduction, such as the coral color chart and the Diving-PAM (for those who are not familiar with these techniques).

6. PLOS authors have the option to publish the peer review history of their article (what does this mean?). If published, this will include your full peer review and any attached files.

Reviewer #1: No

Reviewer #2: **Yes: **Jeremie Vidal-Dupiol

Reviewer #3: No

Reviewer #4: No

---

## [Author Response · Author response to Decision Letter 0]

19 Dec 2023

Reviewer #1: 

- This study helps to improve our knowledge on the disparity of coral colony responses to marine heat waves, why within the same species, certain colonies resist better than others to heat stress? This manuscript provides crucial information allowing us to improve the forecasts about the future of coral reefs but also new useful knowledge to try to establish lines of resistant corals. In this manuscript, the results showing that the few bleached colonies that survived acquired the same ITS2 profiles of the unbleached resistant colonies are extremely interesting, they confirm the essential role of Symbiodiniaceae in resistance/resilience to heat stress and these results are essential to guide future protective measures. I also appreciated the honesty of the authors, who recognize that they cannot decipher if the bleaching was due to seawater warming or freshwater input and do not rule out any possibility. 

The conclusions of this manuscript are important and innovative so I recommend it for publication, I only have minor corrections/suggestions listed below 

REPLY: We thank Reviewer #1 for supporting our work and we addressed the minor comments below.

Abstract 

- Cladocopium spp. 

REPLY: The text was corrected as suggested.

Introduction 

- L46 “Corals experiencing bleaching are particularly vulnerable since they lose up to 90% of the Symbiodiniaceae photosynthate carbon needed for the daily metabolic requirements (9)” Since the work of Muscatine, which now dates back to 1969!, other more recent works have been carried out , estimating in a more precise way the autotrophic C contribution to coral needs in case of bleaching e.g. Tremblay et al 2012 (in Mar. Ecol. Prog. Ser.)

REPLY:The reference suggested was added to the MS

- L84 the authors compared healthy and bleached corals from Bouraké with healthy colonies from a reference site but did they observe also some bleached colonies at the reference site? (even if it is in lower proportions than in Bouraké)

REPLY: No signs of bleaching were observed in A. muricata inhabiting the reference site as wrote in L xx “No coral bleaching was observed at the reference site R1 and, more generally, in the same geographical area”.

- L91 This sentence should be moved to the discussion section. 

REPLY: The sentence was removed as suggested.

Methods 

- L95 Are light and UV data available for the different sites?

REPLY: We agree that it would be interesting to see if a greater light penetration could be one triggering factor for the observed bleaching in Bouraké. Unfortunately, no data on light and UV from the study sites were recorded before the period of bleaching, and according to the slightly lower visibility observed in Bouraké, this would not be the case.

- L124 The authors should add some data on the presence of this species in both Bouraké and reference sites (recovery percentage or others...). This way, it might be easier to consider the consequences on a larger scale of what has been observed here in this study.

REPLY: We appreciate this comment. However, the main goal of this study was to describe the physiological responses of A. muricata and possible mechanisms of bleaching resistance. The data on corals species presence and recovery over time may be more suitable for a new manuscript exploring community evolution over time under the effect of La Niña.

- L144 Wouldn’t it be more appropriate to speak about growth rates instead of calcification rates when the buoyant weight technique is used? We usually consider that alkalinity anomaly technique and 45Ca labelling allow to estimate the calcification rates. 

REPLY: We agree, the term “calcification” was changed in “growth rate”.

For the lipid measurements, is the reference 48 right?? [Folch et al (1956) A simple technique to rule out occlusion of right coronary artery aaer aortic valve surgery. Biol. Chem. 226,497- 509] 

REPLY: Many thanks, the reference was changed with the right one.

Discussion 

- The discussion is clear and well written, the introductory paragraph summarizes the main findings of the paper well. The observed changes in ITS2 are well related to what has been observed in previous works (Camp et al ) and are completely consistent with what has been obtained previously . The results show for the first time remarkably the ability of ITS2 sequences (C1, C1b, C1h) to allow a better resistance/resilience to heat stress. L482 the reference of Reynaud et al (2003) Interacting effects of CO2 partial pressure and temperature on photosynthesis and calcification in a scleractinian corals might be helpful for this part of the discussion

REPLY: Many thanks. We were happy that our data were consistent with previous findings. This strengthens our discussion on the potential role of the symbiotic species in facilitating the coral resistance. The reference was added as suggested.

- L540 “Heterotrophy has already been suggested to be a potential advantage for Bouraké’s corals due to the high loading of organic carbon content released” do we have any idea of the fluctuations of organic matter inside Bouraké through the year? 

REPLY: Heterotrophy has been suggested by Camp et al (2017) as a potential advantage for coral to face the extreme conditions in Bouraké. Maggioni et al. (2021) showed that organic matter daily fluctuated with the tidal cycle, where the concentration of organic matter increases at low tide under the contribution of organic matter released by the mangrove mud. Unfortunately, no data are available throughout the year, and a PhD student is working on the question. Although we still believe that heterotrophy has an important role in the success of corals in Bouraké. We did not specifically measure the heterotrophic ability in this study and, as suggested by Rev 4, our data do not allow to conclude on that. Therefore, we avoid any speculation about heterotrophy in the revised version. 

- Again the conclusion makes a very nice recap of the main findings

Just on comment L559 what do the authors mean by “extreme corals” the notion is not clear 

REPLY: We thank Rev 1. The terms “extreme corals” or “extreme coral communities” are becoming a notion quite common to indicate corals living in (and resisting to) abnormal environmental conditions. The recent review of Schoepf et al, (2023) ‘Coral at the edge of environmental limits: A new conceptual framework to re-define marginal and extreme coral communities’ defined well the concept behind. We added the definition of the term and the reference in the introduction section of the ms. 

 

Reviewer #2:

- In the article entitled “Intra-clade diversity provides resistance to coral bleaching under extreme environmental conditions in Acropora muricata” Alessi et al present the results of an integrative study aiming at understanding potential mechanisms underlying the physiological plasticity of corals that live in extreme and fluctuating conditions during a period of acute stress. To address this question authors performed series of physiological characterization on the cnidarian host and Symbiodiniaceae symbiont on 15 bleached and 15 unbleached corals from an “extreme” site and 15 unbleached corals from a reference site. These data set were analyzed with appropriate bioinformatics and statistical methods and the results obtained are very interesting, welle presented and well discuss. 

My conclusion is that this article deserves a publication in Plos One but see my minor comments below. 

REPLY: We are thankful to Dr Vidal-Dupiol for his comments and suggestions.

Minor comment: 

- I think that the title should be moderate a little bit since a functional link between the symbiodiniaceae community composition and the phenotypic effect was not demonstrated. The term associated rather than provides would be used. 

REPLY: We agree that we have not experimentally demonstrated that the specific association provided bleaching resistance with a certain species of algal symbiont, although our observations suggest it. Accordingly, the title was changed to “Algal symbiont diversity in Acropora muricata from the extreme reef of Bouraké associated with resistance to coral bleaching.”

- Line 47 the sentence is a little bit confusing. Do it mean that the symbiont translocate up to 90% of its photosynthate, or that the symbiont cover up to 90% of the daily carbon needed or that because of the drastic reduction in zoox density the quantity of photosynthate carbon translocated decrease up to 90%? 

REPLY: We agree, the sentence was not clear. We wanted to say that the symbiont cover up to 90% of …. 

The text was changed in “Corals experiencing bleaching are particularly vulnerable since they cannot account on the Symbiodiniaceae photosynthetic carbon needed for the daily metabolic requirements” to avoid confusion.

- Line 249 no segments were underlined on the primers presented line 246-248 

REPLY: The segment was underlined as requested.

- Line 256 precise sequencing depth 

REPLY: Done.

- Line 372 It is important to provide basic sequencing metrics to show that the amount of sequence is enough to support the results. You can either provide a rarefaction curve in suppfile or simply a number of sequence. The out table or equivalent is also classically provided for metabarcoding experiments. Finally you must deposit your sequence data (FastQ files) on a public database (SRA for example). 

REPLY: Many thanks. We reported a rarefaction curve as a supplementary figure (Fig. S3) as suggested, and we made the data public through the suggested SRA (All raw sequence data as fastq read files are accessible under NCBI Sequence Read Archive (SRA), under NCBI's BioProject: PRJNA1020910). 

- I don’t understand how you can link the survival of the 3 BB colonies with the change in ITS2 profile type. What is the cause what is the consequence? Do the stress have induce the change and will make the surviving BB colonies “BZ” colonies the next time, or is the change prior to the survivor to the stress event studied? For me it is quite difficult to conclude but; 1 the difference between the ITS2 profile type of the BZ and RZ at T1 and 2 the similarities between the ITS2 profile type of the BB and RZ at T1 is enough to support your conclusion. The similarities between BB and BZ at T2 is in my opinion the results of a post stress colonization enabled by free space as it was previously shown in other coral species ( for example https://doi.org/10.1111/gcb.12706)

REPLY: This comment agrees with a comment from Rev 3 and is very pertinent. Our data show that bleached and unbleached corals have different ITS2 profile types. While we lacked the profile type of the BB colonies before the bleaching, the BZ colonies were healthy (with a high symbiont density) with an ITS2 profile type completely different from the reference colonies RZ. As Rev 2 said, this already suggests that the ITS2 profiles of BZ are associated with a certain resistance of corals, allowing for their survival. Concerning the three colonies BB that completely recovered a total Symbiodiniaceae density (and metabolic functions), mentioning the Bleaching Adaptive Hypothesis, these three colonies can contribute to the highly debated hypothesis since they acquired/developed the same ITS2 profile type than the resistant BZ colonies. 

What Rev 2 suggested (bleaching post-colonization from opportunistic species) could be a possibility. Unfortunately, this is something we cannot decipher, although it is not fundamental to the goal of this study. 

- Line 562 I don’t understand how it can support adaptation (sensus heritable phenotypic trait). Do you suggest that there is a heritable basis behind the ability to change or not the symbiont community or the heterotrophic capability? 

REPLY: We agree and deleted the mention of adaptation (and heterotrophic capability). We believe that the specific Symbiodiniaceae community we found can support long-term acclimatization, allowing corals to persist in such harsh conditions.

Congratulation for this very interesting work 

Jeremie Vidal-Dupiol 

note: I have recently decided to sign all my reviews because I believe the reviewers should be responsible for their review just as authors are responsible for their article. 

Thank you

 

Reviewer #3: 

- This manuscript focuses on the thermal resilience of corals in New Caledonia and the links to symbiont community composition. The manuscript reports some interesting observations that suggest that a shift in the symbiont community composition, combined with internal lipid reserves, may facilitate survival under increased seawater temperatures, adding to an existing wealth of literature on this topic. Ultimately, I think that the manuscript will be acceptable, but I do have several major comments: 

1) The manuscript contains a very large number of grammatical errors. I appreciate that the primary authors are not native English speakers, but I do note that one of the authors is - can you please make use of this fact and correct the English. I haven't listed all of the errors here (only some are included in my list below) as to do so would take far too long. 

REPLY: Many thanks. The manuscript was checked, and grammatical errors were corrected. We really appreciated the time invested by Rev 3 in revising and adding detailed corrections to our ms.

- 2) It's nice to see the authors use the old CZAR model, however they have not used it correctly. The crucial distinction between the CZAR and simply using a P:R ratio, is that the CZAR includes an estimation or measurement of photosynthate translocation from the symbionts to host. The authors don't use such a metric, so all they're really doing is presenting a daily P:R. Please adjust the manuscript accordingly. 

REPLY: We applied the CZAR formula found in McLachlan et al (2022; see their supplementary information for a detailed description). The same calculation was in Grottoli et al (2006). We agree with Rev 3, removed the CZAR from the manuscript, and used the Pg:R ratio instead. Please note that this change did not affect results.

- 3) This may or may not need changing, but the authors should double-check their Symbiodiniaceae nomenclature, and in particular how to refer to the various Cladocopium types. ITS2 sequence names can still be used where species names are not currently available, but there is convention around how these are stated - they might want to refer to this style guide: https://www.thelifeaquatic.net/?page_id=292

Reply: Many thanks, all in this field is changing too fast! We think we changed all the Symbiodiniaceae nomenclature in the ms accordingly. 

- 4) Of most concern is the impression given by the authors that symbiont 'switching' might be occurring in their study, yet no evidence is provided for this. It's much more likely that the shifts in symbiont community composition resulted from 'shuffling', with cryptic symbiont types in the existing symbiont communities proliferating post-bleaching, rather than de novo uptake of symbionts from the environment. That still might have happened, but there is so little evidence for this in the literature that great caution should be exercised when direct evidence cannot be provided. 

Reply: We fully understand the Reviewer #3 concern. We agree that we cannot decipher if BB colonies that survived and recovered shifted in the symbiont community by either switching to others from seawater or shuffling with cryptic ones. We suggested the first mechanism because we did not observe the presence of the species Cladocopium sub-type C1 in 13 of the BB colonies during the bleaching (only 2 showed a very little percentage of them, see Fig. 5A). However, we acknowledge that the sampling was performed while the colonies were in bleaching, not allowing to be 100% sure in our conclusion. Still, we should have been able to detect a cryptic species during the bleaching since species competition was reduced. As highlighted by Rev 2, the important finding is that BZ colonies did not bleach and had a totally different algal symbiont community than those that bleached (BB). Also, the few BB colonies that survived and recovered from the stress had the same algal symbiont community as BZ. These are, in our opinion, already exciting observations. We attenuated our conclusion, and additional possible explanations for the results obtained were added to the manuscript, where the symbiont switch hypothesis was kept among the others. 

Other comments 

1) Line 57: The wording "by exchanging their microalgal symbiont" is too strong, as it implies symbiont switching (see my comment above). The sentence was changed in “by shuffling their microalgal symbiont”.

2) Line 64: The holobiont also includes the microbiome, so adjust the definition here. The word holobiont was removed.

3) Line 71: "Acroporidae" shouldn't be italicized. Done

4) Line 74: Better as "when a mass bleaching event impacted entire..." The sentence was changed as suggested.

5) Line 79: Better as "The aim of this study was therefore..." The sentence was changed as suggested.

6) line 86: Symbiodiniaceae should start with a capital letter. Done

7) Line 100: Species composition of what exactly? Corals? The word coral was added.

8) Line 104: What are the units for the DO values? Units were added as requested.

9) Line 120: Better as "...four main categories: healthy coral cover..." The text was changed as suggested.

10) Line 141 (and several other places): The correct word is "weighed" rather than "weighted" The text was changed throughout the ms. 

11) Line 147: Why was coral mortality so high? We assumed that BB corals were already under stress and the additional stress of creating nubbing may have cause mortaliny. 

12) Line 149: Should be "to weigh fragments on a..." The text was changed as suggested.

13) Line 163: The specific PAM settings need to be stated here. Done

14) Line 205: What is meant by "sub-surface area"? This whole sentence is unclear. The sentence was removed because unnecessary. Methods description was improved also by adding an additional reference that better describes all the analysis steps.

15) Line 210 and elsewhere: All centrifuge speeds should be stated in g rather than rpm. Many thanks, we did a mistake in reporting the right measure unit (see Tanvet et al. 2023 Ecol and Evolution, for instance). We corrected it. 

16) Line 222: Why were only 3 replicate cell counts measured? This is far too few - it's widely recognised that 8-10 replicate counts are needed for optimal accuracy. Many thanks, we have shortened the sentence too much and lost the exact meaning. In fact, we used three hemocytometers for each coral, reading eight chambers for each, for a total of 24 values (6 replicates). We corrected it.

17) Line 225: Should be "...10 ml of pure acetone WERE added". The text was changed as suggested.

18) Line 234: Better as "normalized to surface area." The text was changed as suggested.

19) Line 243: Unit needs to be superscripted. The text was changed as suggested.

20) Line 278 (and elsewhere): Should be "When significant differences..." The text was changed and the ms checked.

21) Line 289: Should this say "PERMANOVA and pairwise testing were..."? Yes, we changed accordingly.

22) Line 291: Should be "ANOVA was performed with....whereas PERMANOVA pairwise and..." The text was changed as suggested.

23) line 305: Should be "...since the year 1981." The text was changed as suggested.

24) Line 317: Better as "...no signs of polyp extroversion but the skeleton was still white and the coral not fully repopulated by algae" The text was changed as suggested.

25) Line 318: Should this say "old dead CORALS"? The text was changed as suggested.

26) Line 356: Better as "SOME of the initial colonies..." The text was changed as suggested.

27) Line 374 (and elsewhere0: Should be "THE data plot..." The text was changed as suggested and MS checked.

28) Line 384 (and elsewhere): Should be "...more resilient TYPES". Be careful not to confuse the sequence nomenclature with the actual organism. The text was changed as suggested and MS checked. 

29) Line 388: Should be ""more DIVERSE compared..." The text was changed as suggested.

30) Line 399: Acroporidae shouldn't be italicized. The text was changed as suggested.

31) Line 423: Better as "which were 93.82% different..." The text was changed as suggested.

32) Line 427: Should be "which resulted in them being identical..." The text was changed as suggested.

33) Line 437: Better as "IN the Symbiodiniaceae..." The text was changed as suggested.

34) Line 444: Should be "due TO the" The text was changed as suggested.

35) Line 448: Should be "reported IN corals" The text was changed as suggested.

36) Line 451: Should this say "Consistent with Davies et al."? The text was changed as suggested.

37) Line 453: Better as "have greater capacity to..." The text was changed as suggested.

38) Line 454: Should be "traitS" The text was changed as suggested.

39) Line 481: This sentence is a bit vague - in what way was calcification "affected"? The sentence was changed in: “decreased drastically when corals were exposed to combined effect of high pCO2 and high temperatures (31°C)”.

40) Line 487 (and elsewhere): either say "photosynthate" only or "photosynthetic carbon". The text was changed as suggested and MS checked.

41) Line 495: Re-word as "ability was insufficient to cope with..." The text was changed as suggested.

42) Line 506: Should be "recovered FROM the stress" The text was changed as suggested.

43) Line 510: Better as "retained a significantly higher..." The text was changed as suggested.

44) Line 523: Should be "lipid content" The text was changed as suggested.

45) Line 529: What does "all more performant" mean? The sentence was changed in: “the greater concentration in BZ during the bleaching could be due to the improved photo-physiological traits of the specific ITS2 community compared to the reference colonies”.

46) Line 541: Better as "high load of organic carbon released..." The text was changed as suggested.

47) Line 552: Should be "...to cope WITH and evolve..." The text was changed as suggested.

48) Line 556: Should be "...UNDER the combined effect..." The text was changed as suggested.

49) Line 560: Should be "compensate FOR the lack..." The text was changed as suggested.

50) Lines 564-565: Here is another example of overstating the potential for symbiont switching rather than symbiont shuffling. The sentence was changed as follow: “quickly uptake or shuffle Symbiodiniaceae”.

 

Reviewer #4: 

- Alessi et al. observed a coral bleaching event taking place in Bourake lagoon in New Caledonia, an extreme reef exposed to highly variable environmental conditions. The sampled Acropora muricata colonies during and post-bleaching to monitor changes in symbiont community diversity, energetic stores, and various physiological parameters. The found that nearly all colonies that bleached shared the same heat-sensitive symbiont species, whereas nearly all colonies that did not bleach shared a different, putatively heat-tolerant symbiont species. Only three bleached colonies survived to the post-bleaching period, and all had transitioned to the alternate heat-tolerant symbiont species. Meanwhile, analyses of lipids, carbohydrates, lipids, proteins, biomass, photophysiology, symbiont density, chlorophyll, calcification, photosynthesis, and respiration all indicated distinct physiological characteristics for bleached vs. non-bleached vs. recovered vs. control colonies, as well as differences between colonies hosting different symbiont species. Based on these data, the authors conclude that variation in bleaching phenotypes is linked to symbiont community, which in turn influences recovery capacity among corals. 

This experiment was competently performed. I don’t have any major concerns with the design or execution, although of course it is unfortunate that only three colonies survived in one of the treatments, which limits the scope of inference. The ecophysiological methods were all standard for coral studies and the statistics seemed appropriate. Overall, I feel the manuscript could be acceptable after a revision to address some of my concerns about the writing. 

Reply: We are thankful for your support of our work and for your pertinent suggestions. Reviewer 4 is correct, but if, on the one hand, it was unfortunate that only three colonies survived, on the other hand, we were lucky that at least three colonies survived and changed in symbiotic composition. This is an experiment in the wild, and the chance to lose all was high.

- My issues all relate to the narrative. For one, there are dozens of studies like this with similar findings, where researchers observed a bleaching event and took samples to track coral physiology and symbiont dynamics during and post-bleaching, and found that different colonies responded differently, mediated in part by their symbiont communities. The authors, in my opinion, have not done a good job clarifying what makes this study unique, beyond the fact that it was a bleaching event in a particular lagoon in New Caledonia. Based on the title and introduction, I believe the important hook is that the Bourake site is considered an “extreme” reef, exposed to highly variable temperature, pH, dissolved oxygen, and salinity (which was probably more variable than usual thanks to intense La Nina rains during the study period). However, because this is just one extreme reef, it’s hard to draw conclusions about extreme reefs in general, rather than Bourake in particular. The authors don’t really tie their results back to the extreme nature of the reef, possibly because they didn’t measure many of the relevant variables directly, so this angle of the story loses focus. Instead, the manuscript starts to feel generic, often times simply reviewing what has been determined in other, similar studies, and noting that the results here are consistent. If the manuscript were a bit shorter and placed greater emphasis on how the extreme reef conditions may have shaped results, I think it would make a clearer novel contribution. 

Reply: We thank Rev 4, and we partially agree with this comment. We did not sufficiently put the discussion of our findings in the context of the “special” site. We changed a part of the discussion and tried to tie up our results to our study site. However, it is incorrect to say that this was done because we did not measure many relevant variables directly. The ms show a decent representation of the main environmental parameters, which variability has already been shown to be recurrent and regular (see Camp et al. 2017, or better Maggioni et al. 2021). The point is that the main drivers of what we found were temperature and likely salinity. The former was fully measured over the experiment; the latter was extrapolated from the local rain forecast, as usually done in the literature. As Rev 4 suggested, this study is different from the dozens we cited demonstrating differences in the algal symbiont community during and after bleaching because we had the opportunity to observe the event in an extreme reef, which is expected to have corals more likely able to counteract such a harsh condition. This is the unicity of our study, and we hope we made it better in the revised version.

- Moreover, because the reference site also experienced intense rainfall and the sample size for recovered colonies was only three, it’s hard to be confident that the results reported here are truly representative, even of Bourake. That would be ok if the story were framed in a narrow focus, but instead the authors seem to be drawing broad conclusions about different mechanisms of thermal tolerance, from the order in which different sources of energy are being used (even though there were only two time points), to the types of symbionts present (even though colonies were already bleaching when sampled and only three bleached colonies survived), to heterotrophy (not measured), to adaptation. Unfortunately, the authors were limited because they had to build a study around an ongoing bleaching event. Some of the questions could have been better addressed via more regular sampling or aquarium experiments. These limitations could have been better acknowledged, and some of the more speculative conclusions (e.g. about heterotrophy) could have been avoided. Of course, the flip side of the argument is that these are highly ecologically-relevant observations, and creating an extreme reef in an aquarium would be incredibly challenging. 

Reply: Respectfully, we do not base our ms and the discussion on the three colonies only. As remembered by Reviewer 2, the fact that we found differences in the algal symbiont communities between BZ and RZ, provides evidence that unbleached corals in Bouraké have specific symbionts. 100% of the BB colonies that recovered (yes, n=3) acquired (somehow) the same community as the BZ colonies. 

With regard to heterotrophy, we totally agree, and we removed these “speculations” on heterotrophic plasticity. 

Reviewer 4 perfectly understood the dilemma of this study: being highly ecologically relevant without the possibility of demonstrating the mechanism involved because it is simply impossible. The regular sampling evoked by Rev 4 would only have been possible if we had foreseen the event in advance. Concerning the aquarium experiment, our results on the physiological response are coherent with what was reported in our previous studies performed at the same site, suggesting that our results, even if taken in only two-time points (see also Jung et al 2021 for a similar study), should be representative of Bouraké. 

We would like to underline that this is the first time that the energy reserves in tissue are described in Bouraké. These measurements have an important role in the recovery processes of corals (Grottoli et al, 2004). We did our best to improve the discussion and better put our findings in the context of studies performed at our and other extreme sites.

- Another major issue with the writing relates to how the Symbiodiniaceae are described. After the 2018 revision by LaJeunesse et al., the field has largely moved past referring to different symbiont groups as “clades,” which are now considered genera with their own names. The authors strangely combine both clade and genus terminology, at one point talking about “Symbiodinium Clade D” instead of “Durusdinium” but at another talking about “Cladocopium” instead of “Symbiodinium Clade C.” The title itself is misleading: when I read “intra-clade diversity provides resistance to coral bleaching” I honestly though the authors were talking about diversity within a coral clade. Instead, they really mean “intra-generic diversity among coral symbionts provides resistance to coral bleaching.” 

Reply: Many thanks. We agreed and tried to be more consistent in using the correct terminology. However, it is common to find published articles that still report the old nomenclature, even if published after 2018. 

The title was changed to “Algal symbiont diversity…” and we kept only the genus nomenclature throughout the ms.

- From looking at the ITS2 profiles in Figure 5 it’s clear that there are only two main symbiont species in the BZ and BB corals (with a few rarer species in some colonies). I feel the discussion could be simplified if the authors referred to them as two species or, if they prefer, phylotypes. However, it’s wrong to speak as if each sequence variant (C1, C1b, C1c) is a separate entity (as is done in line 445, for example), given that the ITS2 array for a single species contains multiple intragenomic variants in specific proportions. A few other statements betray an older view of symbiont diversity. For example, L58 states that “some symbiont clades (i.e., Symbiodinium clade D) have been described as more resilient than others under elevated temperature.” But the whole group isn’t more heat tolerant, just some species within it. Better to say “some symbiont species (e.g. Durusdinium trenchii) are more resilient than others under elevated temperature.” Another example is L384: “suggesting an early uptake of more resilient sequences.” Corals don’t uptake sequences, they uptake symbionts. I know I’m being nitpicky, but these details are important to get right, otherwise confusion spreads. I’d highly recommend browsing the review by Davies et al. (2023) in PeerJ for a summary of the state of the art in our understanding of Symbiodiniaceae diversity. 

Reply: Many thanks for this comment. We recognize that we are not fully comfortable in that field of research, and we recognize that we have been somewhat induced to the error by the already confused literature on the Symbiodiniaceae nomenclature. We did our best to adjust the discussion.

Other minor concerns: 

- I’d recommend adding + or – symbols in front of percentages when they are reported in the text. Done.

- “Zooxanthellate” is a term for a species characteristic (e.g. reef-building corals are zooxanthellate while deep sea corals are usually azooxanthellate), so I find it strange to refer to non-bleached colonies as zooxanthellate (instead, perhaps call them “healthy” or “non-bleached” or “recovered” depending on the context). Right. While we maintained for convenience the labelling BB, BZ and RZ, we reported BZ and RZ as non-bleached as suggested.

- Some of the figures don’t really need color and might be clearer if depicted in grayscale. Certain elements of the methods could use more introduction, such as the coral color chart and the Diving-PAM (for those who are not familiar with these techniques). Done.

---

## [Editor Report · Decision Letter 1]

22 Dec 2023

Algal symbiont diversity in Acropora muricata from the extreme reef of Bouraké associated with resistance to coral bleaching

PONE-D-23-04677R1

Dear Dr. alessi,

We’re pleased to inform you that your manuscript has been judged scientifically suitable for publication and will be formally accepted for publication once it meets all outstanding technical requirements.

Kind regards,

Anderson B. Mayfield, Ph.D.

Academic Editor

PLOS ONE

Additional Editor Comments (optional):

The fact that PLoS ONE "lost" this article for 3-4 months is, frankly, unacceptable for a major journal, and I hope someone at PLoS ONE is reading this and can provide some sort of explanation. I really hope this sort of oversight by a major publication entity is rare, and I apologize to the authors on behalf of PLoS ONE for their poor handling of the article.
---

## [Editor Report · Acceptance letter]

1 Feb 2024

PONE-D-23-04677R1 

PLOS ONE

Dear Dr. alessi, 

I'm pleased to inform you that your manuscript has been deemed suitable for publication in PLOS ONE. Congratulations! Your manuscript is now being handed over to our production team.

Kind regards, 

on behalf of

Dr. Anderson B. Mayfield 

Academic Editor

PLOS ONE